# PBRL with Offline Reward Modeling on Explicitly Constrained Policies

**Yinglun Xu**[1], **Tarun Suresh**[1], **Rohan Gumaste**[1], **David Zhu**[1], **Ruirui Li**[2],
**Zhengyang Wang**[2], **Haoming Jiang**[2], **Xianfeng Tang**[2], **Qingyu Yin**[2],
**Monica Xiao Cheng**[2], **Qi Zeng**[1], **Chao Zhang**[3], **Gagandeep Singh**[1]
[1]**University of Illinois Urbana-Champaign**, [2]**Amazon**, [3]**Georgia institute of technology**

**Reviewed on OpenReview:** `https://openreview.net/forum?id=LxPg5GJuY3`

## Abstract

Preference-based reinforcement learning (PBRL) in the offline setting has succeeded greatly in industrial applications such as chatbots. A two-step learning framework that learns a reward model from an offline dataset first and then optimizes a policy over the learned reward model through online reinforcement learning has been widely adopted. However, such a method faces challenges from the risk of reward hacking and the complexity of reinforcement learning. To overcome the challenge, our insight is that both challenges come from the state-actions not supported in the dataset. Such state-actions are unreliable and increase the complexity of the reinforcement learning problem. Based on the insight, we develop a novel two-step learning method called PRC: preference-based reinforcement learning on explicitly constrained policies. The high-level idea is to limit the reinforcement learning agent to optimize over policies supported on an explicitly constrained action space that excludes the out-of-distribution state-actions. We empirically verify that our method has high learning efficiency on various datasets in robotic control environments.

## 1 Introduction

Deep reinforcement learning (DRL) is a fundamental learning paradigm for solving sequential decision-making problems and has rich real-world applications (Sutton & Barto, 2018; Luong et al., 2019; Haydari & Yılmaz, 2020). However, traditional DRL approaches require numerical reward feedback during learning, which might be hard to design or obtain in practice. In contrast, binary preference feedback is usually more accessible (Wirth & Fürnkranz, 2013; Wirth et al., 2017; Fürnkranz et al., 2012; Akrour et al., 2012). For example, to judge how well a robot dances, it is challenging to rate each move in a dance. However, it is relatively easier to tell if a piece of dance is better than another. As a result, preference-based reinforcement learning (PBRL) that only requires preference feedback has become a more realistic learning paradigm and is applied in real-world applications (Christiano et al., 2017). Furthermore, as a special instance of PBRL, reinforcement learning from human feedback (RLHF), which utilizes human preferences, has drawn considerable attention and achieved significant success in many NLP tasks (Ouyang et al., 2022; Rafailov et al., 2023). In this work, we focus on PBRL in the offline setting (Zhan et al., 2023; Zhu et al., 2023) where the agent trains on a pre-collected preference dataset on pairs of trajectories. Offline learning is considered to be more practical as it is usually safer and more convenient to access an offline dataset (Levine et al., 2020; Prudencio et al., 2023; Cheng et al., 2022). Particularly, we are interested in a popular learning framework that is widely applied in various PBRL literature and industrial applications, which we call 'two-step learning.'

**Two-step learning framework:** In a typical two-step learning framework (Ibarz et al., 2018; Christiano et al., 2017), the first learning step is reward modeling on an offline preference dataset. The agent approximates the reward model of the environment that can best explain the observations from the training preference dataset. The second learning step is policy optimization, where the learning agent optimizes a

policy to achieve a high reward on the reward model learned in the previous step. This step is typically implemented by standard online reinforcement learning algorithms such as PPO (Schulman et al., 2017). There are multiple reasons behind the vast popularity of the two-step learning paradigm: 1. similar to the motivation of inverse reinforcement learning (Arora & Doshi, 2021), the learned reward model provides a succinct and transferable definition of the task; 2. the method is modular in that each phase of learning can be implemented by existing well-developed methods. Despite its popularity, this method faces two main challenges for offline PBRL.

- **Reward over-optimization/Pessimistic learning:** Pessimistic learning has always been the key challenge in offline learning settings. Given a policy, due to the distribution mismatch between the trajectory distribution of the dataset and that induced by the policy, the agent cannot accurately evaluate the policy if the mismatch is large. Reward over-optimization can happen if an agent overestimates and outputs a policy that achieves a high score on the learned reward model but has poor performance in the real environment.

- **Reinforcement learning complexity:** Implementing Reinforcement learning efficiently is known to be challenging (Dulac-Arnold et al., 2021). In addition, the stochastic binary preference feedback contains less information compared to the exact and accurate scalar reward feedback. This can influence the quality of the learned reward model and might increase the difficulty of policy optimization. We have empirically verified the difficulty of running an online reinforcement learning process on the reward model learned from preference signals compared to reward signals in our experiments from Section 4.

Most studies on offline PBRL seek other learning frameworks to circumvent the two challenges or increase the learning efficiency from aspects. For example, Hejna & Sadigh (2023); An et al. (2023) consider frameworks that do not learn a reward model. Kim et al. (2023); Zhang et al. (2023) considers using more advanced methods for reward modeling and trajectory generation, and the focus is not on solving the two challenges above. Standard two-step learning methods (Christiano et al., 2017) tackle the first challenge by applying KL regularization during the policy optimization step, but the second challenge from reinforcement learning difficulty remains. Due to the popularity and advantages of the two-step learning framework in practice and its advantages as mentioned before, this work aims at increasing the learning efficiency of the two-step learning framework by solving the two challenges above at the same time.

**Our contributions.** In this work, we propose a novel two-step learning PBRL algorithm called 'PRC' that simultaneously tackles the two main challenges above. Our key insight is that both challenges are induced by the same element: the state actions that are out of the dataset distribution. For pessimistic learning, the agent must avoid the policies that are more likely to visit such state-actions as the agent cannot evaluate their performance accurately. Meanwhile, these out-of-distribution state-actions contribute to the complexity of the learning problem. Therefore, our insight is to constrain the action space to exclude such out-of-distribution state-actions. In this way, our method not only follows the idea of pessimism but also reduces the complexity of the corresponding reinforcement learning problem. In the experiments, we apply our method to the continuous control problems, which are the standard RL benchmark. We construct offline PBRL datasets based on the offline RL benchmark D4RL (Fu et al., 2020). We empirically verify that, compared to most baseline methods, our algorithm always learns a policy of better performance. When compared to a recent state-of-the-art offline PBRL method 'flow to better' (FTB) (Zhang et al., 2023), PRC has similar performance. However, FTB requires access to an additional behavior dataset that is not used by PRC. More importantly, FTB uses the technique of diffusion (Yang et al., 2023), which is computationally expensive. On our benchmark, the PRC algorithm achieves a $7\times$ speedup compared to FTB while achieving similar performance.

## 2 Preliminary

### 2.1 Offline Preference Reinforcement Learning Problem

We consider preference-based reinforcement learning with offline preference feedback (Zhan et al., 2023). The agent has access to an environment characterized by an incomplete Markov Decision Process (MDP)

without a reward function $\mathcal{M} = (\mathcal{S}, \mathcal{A}, \mathcal{P}, \mu_0)$, where $\mathcal{S}$ is the state space, $\mathcal{A}$ is the action space, $\mathcal{P}$ is the state transition function, and $\mu_0$ is the initial state distribution. A policy $\pi : \mathcal{S} \to \Delta(\mathcal{A})$ maps a state $s$ to an action stochastically drawn from $\pi(\cdot|s)$. We denote $\Pi^{\mathcal{S},\mathcal{A}}$ as the class of policies supported on the state space $\mathcal{S}$ and action space $\mathcal{A}$. A trajectory of length $t$ is a sequence of states and actions $(s_1, a_1, \ldots, s_t, a_t, s_{t+1})$ where $a_i \in \mathcal{A}, s_{i+1} \sim P(\cdot|s_i, a_i), \forall i \in [t]$. A preference model $F$ takes a pair of trajectories $\tau_1, \tau_2$ of the same length as input, and outputs a preference $\sigma \in \{\succ, \prec\}$, indicating its preference over the two trajectories, i.e. $\tau_1 \succ \tau_2$ or $\tau_1 \prec \tau_2$. In this work, we assume the standard conventions and consider the Bradley-Terry (BT) family of preference models (Bradley & Terry, 1952). Specifically, there exists a reward function $R : \mathcal{S} \times \mathcal{A} \to \mathbb{R}$, such that the probability distribution of the output preference of $F$ given $\tau_1, \tau_2$ satisfies:

$$F^R(\succ |\tau_1, \tau_2) = \frac{\exp(\sum_{(s,a)\in\tau_1} R(s,a))}{\exp(\sum_{(s,a)\in\tau_1} R(s,a)) + \exp(\sum_{(s,a)\in\tau_2} R(s,a))} \tag{1}$$

We define the performance of a policy $\pi$ on a reward function $R$ as the expected cumulative reward of a trajectory generated by the policy. Formally, we abuse the notation and denote the policy performance as $R(\pi) := \mathbb{E}_{s^0 \sim \mu_0, a^i \sim \pi(s^i), s^{i+1} \sim P(\cdot|s^i, a_i)}[\sum_i \gamma^i \cdot R(s^i, a^i)]$, where $\gamma \in [0, 1]$ is a discount factor. An offline preference dataset $D = \{(\tau_1^1, \tau_2^1, \sigma^1), \ldots, (\tau_1^N, \tau_2^N, \sigma^N)\}$ consists of multiple preference data over different trajectory pairs. We use the term 'behavior policy' to represent how the trajectories are generated in the dataset. The action distribution given by the behavior policy is the same as that of the dataset (Wu et al., 2019). A learning agent is given access to the incomplete MDP $\mathcal{M} = (\mathcal{S}, \mathcal{A}, \mathcal{P})$ of the environment and a dataset $D$ whose preferences are generated by a preference model $F$. The learning goal is to learn a policy that has high performance on the reward function $R$ of the preference model $F$. We say a learning algorithm is efficient if it can learn such a high-performing policy.

## 2.2 Model-Based Two-Step Learning Framework

Here, we introduce a prevalent model-based two-step learning framework widely applied for solving PBRL problems (Christiano et al., 2017) and is the framework for our algorithm in this work. The general algorithm consists of the two learning steps as follows.

1. **Step 1: Offline Reward Modeling.** The agent learns the optimal reward model $\hat{R}$ in $\mathcal{R}$ that minimizes the prediction loss on the preference dataset: $\hat{R} \leftarrow \arg\min_R \mathcal{L}_D^p(R)$.

2. **Step 2: Online Policy Optimization (Online Reinforcement Learning).** The agent learns the optimal policy on the learned reward model in a constrained policy class: $\hat{\pi} \leftarrow \max_{\pi \in \Pi_C} \hat{R}(\pi)$.

Here, The first step is usually solved by a standard supervised learning process. At this step, $\mathcal{L}_D^p(\cdot, \cdot)$ is a prediction loss function that evaluates how well a reward model reproduces the preference labels in the dataset. A standard choice is the sum of the negative log probability of the reward model predicting the correct preference: $\mathcal{L}_D^p(R) := \sum_{(\tau_1, \tau_2, \sigma) \sim D} - \log F^R(\sigma|\tau_1, \tau_2)$. The second step is usually solved by an online reinforcement learning process on the incomplete environment MDP combined with the reward model learned from the first step. At this step, the agent specifies a constrained policy class $\Pi_C \subseteq \Pi^{\mathcal{S},\mathcal{A}}$ to avoid the risk of reward hacking. A standard choice is policies with limited KL divergence to the behavior policy from which the dataset's trajectories are generated (Christiano et al., 2017). This requires the agent to learn a behavior clone policy that approximately reproduces the action distribution in the dataset. To achieve this, the agent specifies an imitation loss function $\mathcal{L}_D^i(\pi)$ that measures how well a policy imitates the behavior in the dataset. Then, the agent learns a behavior clone policy that minimizes the imitation loss $\pi_i \in \arg\min_\pi \mathcal{L}_D^i(\pi)$. This is often achieved by a supervised learning process. Let $\text{KL}(\pi_1, \pi_2) := \mathbb{E}_{\tau \sim \pi_1} \sum_{(s,a)\in\tau} \log \frac{\pi_1(a|s)}{\pi_2(a|s)}$ be the definition of the KL divergence between two policies. Then the constrained policy class can be defined as $\Pi_C := \{\pi : \text{KL}(\pi, \pi^i) \leq \beta\}$, where $\beta > 0$ is a constraint parameter.

In practice, it is hard to directly search for the optimal policy on $\Pi_C$ specified by the KL regularization constraint above. A practical way to solve the problem is to introduce a Lagrange multiplier to the optimization goal and search for the optimal policy in the entire policy space: $\max_\pi \hat{R}(\pi) - \alpha \cdot \text{KL}(\pi, \pi_i)$ (Zhan et al.,

2023), which can be solved by a standard RL algorithm such as PPO (Schulman et al., 2017). Formally, Algorithm 1 below shows a popular practical implementation of two-step learning. The reward model $\hat{R}$ and the imitation policy $\pi_i$ can be learned by standard supervised fine-tuning methods.

---

**Algorithm 1:** Two-step training with KL-regularization and PPO algorithm

**Inputs** : Environment $(\mathcal{S}, \mathcal{A}, \mathcal{P})$, Dataset $D$, prediction loss function $\mathcal{L}_D^p$, imitation loss function $\mathcal{L}_D^i$

1. Train a reward model $\hat{R} \leftarrow \arg\min_{R \in \mathcal{R}} \mathcal{L}_D^p(R)$
2. Train an imitation policy $\pi_i \leftarrow \arg\min_{\pi \in \Pi} \mathcal{L}_D^i(\pi)$
3. Run PPO algorithm to solve the optimization problem $\hat{\pi} \leftarrow \arg\max_{\pi \in \Pi^{\mathcal{S},\mathcal{A}}} R(\pi) - \alpha \cdot \mathrm{KL}(\pi, \pi_i)$
4. Output $\pi^*$

---

# 3 Preference Based Reinforcement Learning on an Explicit Constrained Policies (PRC)

## 3.1 Sub-Optimality of Policy Optimization on Regularized Learned Reward Model

In this work, we focus on the choice of the constrained action space $\Pi_C$ in the policy optimization step. Let $\hat{R}$ be the reward function learned in the reward modeling step, and $R^*$ be the true reward function. We define $\Gamma_{\hat{R}}(\pi, \pi_{\mathrm{ref}}) := |[R^*(\pi) - R^*(\pi_{\mathrm{ref}})] - [\hat{R}(\pi) - \hat{R}(\pi_{\mathrm{ref}})]|$ as the estimation error on a policy $\pi$ given by the reward model $\hat{R}$ compared to a reference policy $\pi_{\mathrm{ref}} \in \Pi^{\mathcal{S},\mathcal{A}}$. We focus on relative rewards because the preference model in Eq. 1 only depends on relative rewards. Intuitively, a policy will have low estimation error if it is likely to visit the state actions covered by the dataset. A rigorous analysis of $\Gamma_{\hat{R}}(\cdot)$ for a reward model $\hat{R}$ learned by deep neural networks based function approximation remains an open question. Based on the definition of $\Gamma_{\hat{R}}(\cdot)$, we can give a theoretical bound on the performance of the policy $\hat{\pi}$ learned at the second step in Proposition 3.1.

**Proposition 3.1.** *Let $R^*$ be the true reward function, $\hat{R}$ be the learned reward function from the dataset, and $\Gamma_{\hat{R}}(\cdot)$ be the relative estimation error of the learned reward $\hat{R}$. Let $\hat{\pi}$ be the learned policy to approximate the optimal policy in $\hat{R}$, and $\epsilon_{opt} := \max_{\pi' \in \Pi_C} \hat{R}(\pi') - \hat{R}(\hat{\pi})$ be the corresponding optimization error. Then, the performance of the learned policy is at least*

$$R^*(\hat{\pi}) \geq \max_{\pi \in \Pi_C, \pi_{ref} \in \Pi^{\mathcal{S},\mathcal{A}}} \left( R^*(\pi) - \Gamma_{\hat{R}}(\pi, \pi_{ref}) - \max_{\pi' \in \Pi_C} \Gamma_{\hat{R}}(\pi', \pi_{ref}) \right) - \epsilon_{opt}.$$

The proof can be found in the appendix. By Proposition 3.1, we can better understand the difficulties of reward hacking and the complexity of reinforcement learning in the two-step learning framework. Here, the last term $\epsilon_{\mathrm{opt}} := \max_{\pi' \in \Pi_C} \hat{R}(\pi') - \hat{R}(\hat{\pi})$ represents the optimization error during the policy optimization process. Intuitively, the optimization error would be large if the policy optimization problem is hard to solve. The first term $\max_{\pi \in \Pi_C, \pi_{\mathrm{ref}} \in \Pi^{\mathcal{S},\mathcal{A}}} \left( R^*(\pi) - \Gamma_{\hat{R}}(\pi, \pi_{\mathrm{ref}}) - \max_{\pi' \in \Pi_C} \Gamma_{\hat{R}}(\pi', \pi_{\mathrm{ref}}) \right)$ implies a trade-off for the choice of the constrained policy class $\Pi_C$. Enlarging the constrained policy space can include policies of higher performance $R^*(\pi)$. However, the maximal estimation error in the constrained policy space $\min_{\pi_{\mathrm{ref}}} \max_{\pi' \in \Pi_C} \Gamma_{\hat{R}}(\pi', \pi_{\mathrm{ref}})$ also increases, which can cause reward hacking. To balance the trade-off, the constrained policy space should be limited to a subset of policies with low estimation error. This idea is closely related to the concept of 'pessimism,' which requires the agent to focus on the policies covered by the dataset.

As shown in Alg 1, an intuitive and popular way to balance the trade-off is to use KL regularization to specify the constrained policy space (Christiano et al., 2017). However, this approach could suffer from the difficulty of RL training for finding the optimal policy on the regularized reward among the full policy class $\pi \in \Pi^{\mathcal{S},\mathcal{A}}$, resulting in significant optimization errors. This is also observed in our experiments in Section 4. Next, we introduce our approach to balance the estimation-constraint error while reducing the optimization error.

### 3.2 Policy Optimization on An Explicitly Constrained Policy Space

A straightforward way to reduce the optimization error is to reduce the complexity of the optimization problem. Our high-level idea is to ensure that the agent can directly search for the optimal policy in the constrained policy class instead of searching among all possible policies. This can be achieved by explicitly choosing specific policies for the constrained policy class. In order to avoid the problem of reward over-optimization, such policies should mostly visit the state-actions covered by the dataset. Therefore, we define a special constrained policy space with an imitation policy $\pi^i$ and a pessimistic coefficient $p$ as below:

$$\Pi_C := \{\pi : \forall(s,a) \text{ s.t. } \pi^i(a|s) < p, \pi(a|s) = 0\}$$

Here, we abuse the notation of $\pi(a|s)$. If the action space is discrete, then $\pi(a|s)$ represents the probability of the policy to generate the response $a$ at the state $s$. If the action space is continuous, the $\pi(a|s)$ becomes the probability density. The constrained policy class $\Pi_C$ consists of policies supported on a constrained action space. Given a value of $p$, the constrained action space at a state $\mathcal{A}'|s$ includes the actions that satisfy $\pi^i(a|s) \geq p$. This choice tackles the problem of reward hacking in a way similar to the KL regularization approach. If the imitation policy $\pi^i$ can represent the action distribution in the dataset, then the constrained policy class here explicitly restricts the agent to consider policies that only take the actions likely covered by the dataset. More importantly, the complexity of finding the optimal policy is reduced when the agent only focuses on a constrained policy space instead of the full policy space. Here, the agent only needs to search among the policies supported on a constrained action space that is much smaller than the original action space. This can make the reinforcement learning process easier and reduce the optimization loss. We use an intuitive example to show how much optimization loss can be reduced. Consider the case of the environment being a tabular MDP where mature theoretical understandings have been developed. For the popular and efficient UCBVI algorithm (Azar et al., 2017), the dependency of optimization loss on the size of action space $|A|$ is $O(\sqrt{|A|})$. In the case of learning on a clipped action space, if the clipped action space has a size of $|A|/N$, then the optimization loss is reduced by $1/\sqrt{N}$ times. In our practical implementations in Section 4, the size of the clipped action is $10^{-d}$ times smaller than that of the full action space, where $d$ is the number of action space dimensions. Intuitively, learning on such a small action space can significantly reduce the optimization error, and we further empirically verified this in Section 4.

Formally, we show the general form of the PRC algorithm using the constrained policy class above in Alg 2. In the first step, the learning agent learns a behavior policy that can best imitate the behavior demonstrated in the dataset and then learns a reward model that best interprets the dataset's preference label. In the second learning step, the agent learns the policy supported on a constrained action space $\mathcal{A}'$ that maximizes the cumulative return on the learned reward model.

---

**Algorithm 2:** Preference Based Reinforcement Learning on a Constrained Action Space

**Inputs** : Environment $(\mathcal{S}, \mathcal{A}, \mathcal{P})$, Dataset $D$
1. Train a reward function $\hat{R} \leftarrow \arg\min_R \mathcal{L}_D^p(R)$.
2. Train a behavior clone policy $\pi_b \leftarrow \mathcal{L}_D^i(\pi)$.
3. Construct a clipped action space $\mathcal{A}'$ such that at a state $s$, the clipped action space is
   $\mathcal{A}'|s = \{a : \pi_b(a|s) \geq p\}$.
4. Find the optimal policy $\hat{\pi}$ supported on the clipped action space $\hat{\pi} \leftarrow \arg\max_{\pi \in \Pi_{\mathcal{A}'}} \hat{R}(\pi)$.
4. Output $\pi^*$

---

### 3.3 Practical Implementation of PRC

Finally, we present a practical implementation of the PRC algorithm for our experiments on continuous control problems. In this case, the action space is continuous and has multiple dimensions. Our goal is to determine the clipped action space that is easy to represent in practice. To achieve this, we train a deterministic imitation policy $\pi^i$ that generates actions close to the ones in the dataset and then specifies the constrained action space centered at the actions given by the policy with a radius $r$. In the context of

Algorithm 2, this process essentially considers a stochastic behavior clone policy $\pi_b(\cdot|s)$ that has a uniform distribution over a constrained action space centered at $\pi_i(\cdot|s)$ with radius $r$. The action given by the deterministic policy can be thought of as an empirical estimation of the center of the distribution given by the behavior policy of the dataset. Since we have no prior knowledge about the shape of the distribution of the behavior policy, we adopt a reasonable assumption that an action is more likely to be sampled by the behavior policy if it is closer to the center of its distribution. Then, we approximate the space of $\mathcal{A}'$ as a box whose center is at the deterministic policy with a radius $r$, which is a hyperparameter related to the degree of pessimism. As we show in Appendix A, the method based on this approach is not sensitive to the value of $r$.

Formally, in Alg 3, we show our practical implementation for the PRC algorithm. Lines 1-2 learn a reward model and an imitation policy, which a standard supervised learning approach can achieve. The loss functions are set in Eq. 2. Here, $d(\cdot, \cdot)$ measures the distance between two actions. In our experiments, we consider the standard continuous control problems, and the distance measure is defined as the L2 distance between the two actions on the action space. Lines 3 construct a constrained action space for each state centered at the action given by the imitation policy. Line 5 learns the optimal policy $\hat{\pi}' : \mathcal{S} \to \mathcal{A}'$ supported on the constrained action space. In practice, we can solve this by applying the SAC or PPO algorithm. Given a state, the output action satisfies $a = \text{Proj}_{\mathcal{A}}(\pi^i(s) + a')$, where $a' \sim \hat{\pi}'(\cdot|s)$. Note that this process is similar to how the batch-constrained Q learning (BCQ) (Fujimoto et al., 2019) algorithm uses the perturbation function in its implementation. While our practical implementation of PRC shares similarities with BCQ, in the appendix, we discuss the fundamental differences between BCQ and PRC in detail.

$$
\begin{aligned}
\mathcal{L}_D^p(R) &= -\sum_{(\tau_1, \tau_2) \in D} \log \frac{\exp(\sum_{(s,a) \in \tau_1} R(s,a))}{\exp(\sum_{(s,a) \in \tau_1} R(s,a)) + \exp(\sum_{(s,a) \in \tau_2} R(s,a))} \\
\mathcal{L}_D^i(\pi) &= \sum_{(s,a) \sim D} d(\pi(s), a)
\end{aligned}
\tag{2}
$$

---

**Algorithm 3:** PRC practical implementation

---

**Inputs**       : Environment $(\mathcal{S}, \mathcal{A}, \mathcal{P}, \mu_0)$, Dataset $D$
**Parameters:** Positive real number $r$
1. Train a reward function $\hat{R} \leftarrow \arg\min_R \mathcal{L}_D^p(R)$
2. Train a deterministic behavior clone policy $\pi^i \leftarrow \mathcal{L}_D^i(\pi)$.
3. Construct a constrained action space $\mathcal{A}' = \mathbb{R}^N$ where $N$ is the dimensions of $\mathcal{A}$. Each dimension is constrained on $[-r, r]$.
4. Find the optimal policy $\hat{\pi}'$ that optimizes $\hat{\pi}' \leftarrow \arg\max_{\pi \in \Pi^{\mathcal{S}, \mathcal{A}'}} \mathbb{E}_{s \sim \mu_0, a' \sim \pi(\cdot|s)} \hat{R}(s, \text{Proj}_{\mathcal{A}}(\pi^i(s) + a'))$
5. Output $\hat{\pi} : \hat{\pi}'(a|s) = \hat{\pi}(\text{Proj}_{\mathcal{A}}(\pi^i(s) + a')|s)$

---

## 4 Experiments

### 4.1 Setup

**Dataset:** Following previous studies (Hejna & Sadigh, 2023; Zhang et al., 2023), we construct our offline preference dataset from D4RL benchmark (Fu et al., 2020) labeled by synthetic preference following the preference model in Section 2. The details of our dataset construction can be found in the Appendix. Note that we consider the offline setting where the agent can only access a dataset of preference labels. We consider various datasets from D4RL to represent the general learning scenarios. The robot control environments we choose include 'Hopper', 'HalfCheetah', and 'Walker'. The types of trajectories we choose include 'Medium', 'Medium-Replay', and 'Medium-Expert'. To make the number of state-action in the dataset aligned with that of the D4RL benchmark, the total number of trajectory pairs with a preference label is $1 \times 10^6$.

**Baseline Algorithms:** Here, we consider multiple two-step learning algorithms and state-of-the-art attacks in the related offline PBRL setting as baselines, which are listed below:

1. Two-step learning with KL-regularization and no regularization: We adopt the traditional two-step learning baseline with and without KL-regularization as introduced in Section 2. To make the comparison fair, the algorithms initialize their policies as the behavior clone policy.

2. Reward modeling followed by standard offline RL algorithms: Another simple yet efficient two-step learning approach for offline PBRL problems is to combine reward modeling with a standard reward-based offline RL algorithm. First, a reward model is learned from the preference dataset. Then, the reward model is used to provide scalar reward labels to the state-action in the dataset. Finally, we apply a standard offline RL algorithm IQL (Kostrikov et al., 2021) training on the dataset with scalar reward labels.

3. Inverse preference learning (IPL) (Hejna & Sadigh, 2023) and flow to better (FTB) (Zhang et al., 2023): IPL and FTB are the state-of-the-art learning algorithms for a related offline PBRL setting that requires a preference dataset and a behavior dataset. To avoid underestimating the performance of them, these algorithms have access to the full D4RL dataset during training. Note that our algorithm has no access to the extra behavior dataset.

4. Oracle: The oracle is trained with true rewards instead of preference labels. Here, we apply IQL training on the base D4RL dataset with true reward signals. Note that the information from the reward signals is strictly more than that from the preference signals in this case. The oracle should be considered as an upper bound on the performance of an offline PBRL algorithm.

To ensure a fair comparison, all the methods that require reward modeling share the same learned reward model during training. Additional detailed training setups can be found in the Appendix.

## 4.2 Learning Efficiency Evaluation

Here, we compare the efficiency of PRC against other baseline algorithms on different datasets. To straightforwardly compare the performance of different learning algorithms, we show the standard D4RL score (normalized performance) of the policy learned by a method during training.

The results in Table 1 show that the PRC algorithm has high learning efficiency. Here, we use the SAC algorithm for the online reinforcement learning step in PRC. The baseline two-step learning methods use the PPO algorithm for the online learning step, as it achieves better performance. PRC is generally more efficient than most baselines and sometimes even competes with the oracle. Our algorithm is a more efficient two-step learning method as it performs much better than other two-step learning baselines initialized from the behavior clone policy. This observation indicates that setting the initial policy as the behavior clone policy is not enough to achieve high learning efficiency, and training on the constrained action space is the key to high learning efficiency for PRC. Even with additional access to the behaviors in the D4RL dataset, PRC algorithm outperforms the IPL algorithm. Although PRC and FTB have similar performance, we notice that FTB uses the diffusion technique (Yang et al., 2023) during training, which is very computationally expensive. In our testing, the training time required by the FTB is about 7 times as long as that required by PRC. The details can be found in the Appendix. In conclusion, the PRC algorithm has high learning efficiency while being computationally efficient.

In addition, we note that the PRC algorithm achieves high learning efficiency on the medium-expert dataset. The action distribution of this dataset is multimodal, contributed by multiple policies. Half of the trajectories in the dataset are collected by a medium policy, while the rest are collected by an expert policy. In principle, a single deterministic behavior clone policy is not able to well characterize a multimodal dataset action distribution, which is a limitation on the implementation. The reason for the empirical success of our method could be that at each state, the behavior clone policy learns to predict an action that is close to the action given by either policy, instead of some actions in the middle that are not close to any dataset policies. In Section 6, we discuss this limitation on our method and the future direction to make it more universal.

Next, we empirically analyze why PRC should be an efficient learning algorithm from the aspect of pessimistic learning and reduced reinforcement learning complexity.

| Dataset | Oracle | PRC | Naive two-step | KL two-step | IPL | RM | FTB |
|---|---|---|---|---|---|---|---|
| HalfCheetah-Medium | 47.3 ± 0.2 | **47.0 ± 0.5** | 40.1 ± 0.5 | 41.9 ± 0.1 | 42.7 ± 0.1 | 43.2 ± 0.1 | 43.5 ± 0.3 |
| HalfCheetah-Medium-Replay | 46.1 ± 0.1 | **43.3 ± 0.2** | 31.9 ± 0.8 | 32.9 ± 0.9 | 34.9 ± 3.1 | 40.1 ± 0.7 | 38.6 ± 1.1 |
| HalfCheetah-Medium-Expert | 92.1 ± 0.7 | 78.4 ± 2.9 | 40.9 ± 0.2 | 40.4 ± 0.5 | 41.7 ± 1.0 | 48.2 ± 0.8 | **86.0 ± 0.5** |
| Hopper-Medium | 76.1 ± 1.2 | **71.0 ± 7.5** | 67.5 ± 2.4 | **73.5 ± 4.0** | **72.4 ± 7.4** | 67.2 ± 0.3 | 66.8 ± 5.7 |
| Hopper-Medium-Replay | 76.7 ± 5.3 | 47.0 ± 19.5 | 49.4 ± 7.4 | 53.1 ± 2.2 | 39.5 ± 17.5 | 32.2 ± 0.4 | **62.9 ± 9.2** |
| Hopper-Medium-Expert | 113.1 ± 0.4 | 100.2 ± 9.5 | 67.3 ± 7.1 | 71.4 ± 10.9 | 76.9 ± 8.1 | 97.1 ± 5.2 | **109 ± 1.7** |
| Walker2d-Medium | 84.5 ± 0.3 | **84.4 ± 0.8** | 81.0 ± 0.8 | 79.5 ± 2.1 | 80.8 ± 1.6 | **81.9 ± 2.5** | 80.5 ± 0.6 |
| Walker2d-Medium-Replay | 83.1 ± 2.3 | **87.9 ± 6.1** | 49.8 ± 7.3 | 45.4 ± 0.1 | 49.3 ± 3.5 | 71.8 ± 6.4 | 79.3 ± 1.9 |
| Walker2d-Medium-Expert | 111.5 ± 0.4 | **110.4 ± 0.6** | 93.4 ± 4.9 | 92.1 ± 2.3 | **107.5 ± 3.0** | **105.7 ± 6.9** | **109 ± 0.3** |

Table 1: Comparison between the performance of different learning methods. Oracle is trained on the true reward signals instead of preference signals, which is not a baseline. Bolded performances are within 95% of the highest performance. FTB requires much more training time than other methods.

## 4.3 Pessimism Effectiveness

Here, we empirically verify that by training on a constrained action space, the PRC algorithm achieves effective pessimistic learning. In Figure 1, we show some representative examples. For the PRC algorithm, during training, the trend of the learned policies' performance measured on the learned reward model is aligned with that measured on the true reward. This suggests that the pessimistic learning in the PRC algorithm is effective: it mainly considers the policies that are supported by the dataset so that the agent can evaluate their relative performance accurately. In contrast, for the KL-regularized two-step learning method, we find multiple cases where the trends can even be opposite when evaluated on the reward model and on the true reward. During training, the simulated performance of the learned policy increases while the actual performance decreases. This is a result of relying on overestimated rewards during policy optimization. Since the methods use the same learned reward model, the results support that our method achieves pessimistic learning effectively by exploiting the more accurate prediction of the learned reward model.

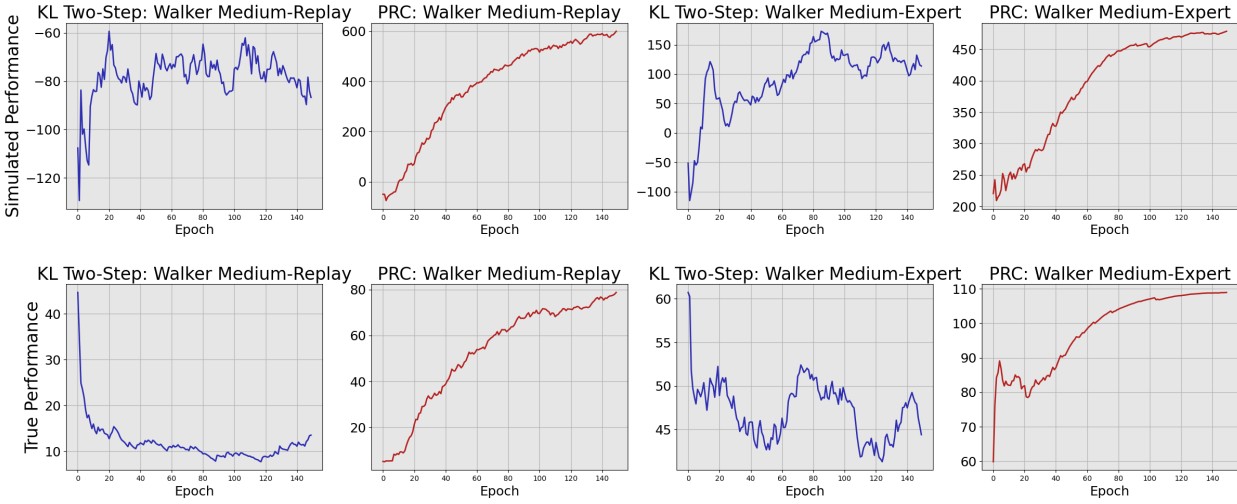

Figure 1: Comparison between the trend of the performance of the learned policies on the learned (simulated performance) and true reward models (true performance) during training. An algorithm is not pessimistic enough if the trends of the simulated and true performance over time are not aligned.

## 4.4 Reinforcement Learning Efficiency on Constrained Action Space

Here, we empirically verify that reinforcement learning is much easier on the constrained action space compared to learning on the full action space. We measure the simulated performances of the policies learned by the PPO algorithm on the full or the constrained action spaces. Therefore, the baseline method is the naive two-step learning that uses PPO for policy optimization, and the PRC method also uses PPO in this case. In Figure 2, we observe that the simulated performance of the policies learned by the PRC method (optimizing over a constrained action space) is much higher than that learned by the naive two-step learning methods (optimizing over the full action space). While the baseline method struggles with low-performing policies, the PRC method steadily learns policies of much higher performance. Considering that the optimal policy supported on a constrained action space is no higher than that supported on the full action space, the empirical results show that the optimization error of reinforcement learning on a constrained action space is much less than that on the full action space. Therefore, the empirical observation verifies our statement that the PRC method has less optimization error during reinforcement learning.

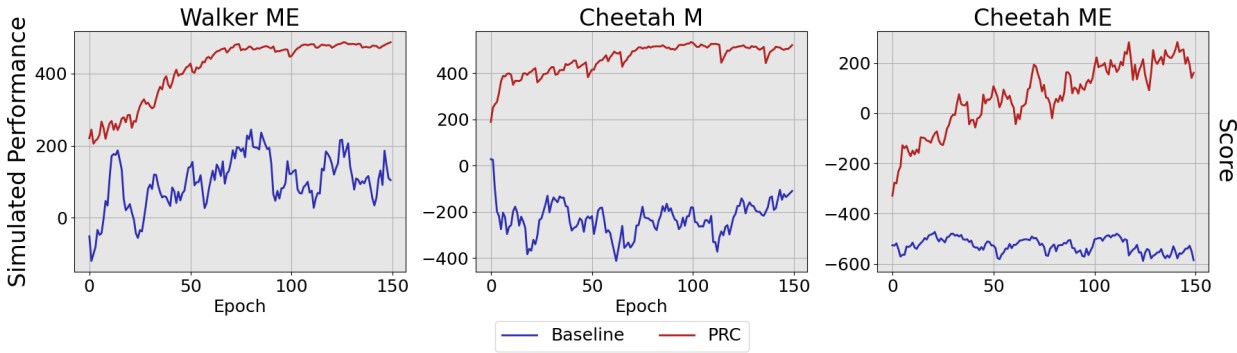

Figure 2: Comparison between the simulated performance of the policies learned by different methods. The baseline method is the naive two-step learning method. The learning efficiency is measured by the simulated performance of the learned policies.

## 4.5 Reinforcement Learning Complexity in PBRL

Here, we empirically show that solving the policy optimization process with a standard RL algorithm on a reward model learned from preference signals is harder than that from true reward signals. Given the same D4RL dataset, we label the dataset with preference signals, as in the main experiments, and true reward signals. Then, we train two reward models based on the two kinds of signals. Finally, we train an efficient RL algorithm, PPO, on the two reward models and compare the learning efficiency in the two cases. The results in Figure 3 show that when learning on the reward model trained from true rewards, the RL agent can quickly learn some policies that have high performance on the reward model. In comparison, it can take much longer for the agent to find a good policy on the reward model from preference signals, and sometimes, the agent may not be able to find a decent policy after a lot of training epochs. This indicates that it is harder to learn from a reward model that is trained on preference signals instead of true reward signals.

In the appendix, we have added additional experimental results for evaluating our method. In Appendix A.1, we test our method on the offline dataset with human preference labels based on the D4RL and Metaworld benchmark (Kim et al., 2023). We show that our method can still learn policies of high performance from human preference feedback. In Appendix A.2, we conduct an ablation study on the constrained action space radius $r$. We show that our method is not sensitive to the choice of this hyperparameter, and it can always learn policies of high performance for different values of $r$.

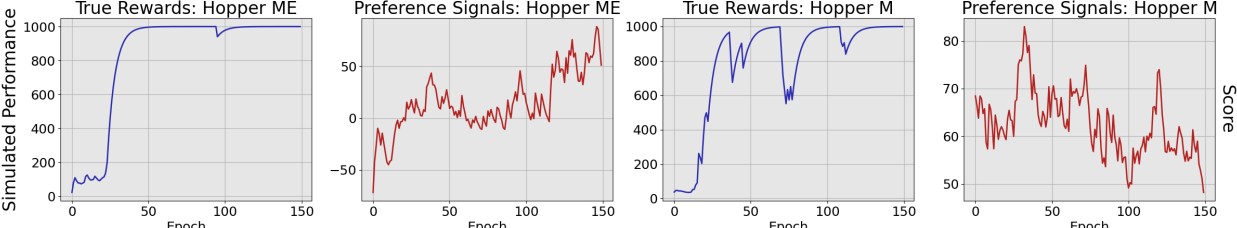

Figure 3: For each dataset, two reward models are trained on the true rewards and preference signals, respectively. The same RL algorithm, PPO, is applied to learn on both reward models. The difficulty of policy optimization on a reward model is large if the PPO algorithm takes more training epochs to find a policy of high performance.

## 5   Related Work

**Offline Reinforcement learning:** There have been many studies on the standard offline reinforcement learning setting with reward feedback (Kumar et al., 2019; Fujimoto et al., 2019; Wu et al., 2019; Levine et al., 2020; Cheng et al., 2022; Yu et al., 2021; Fujimoto & Gu, 2021; Bhardwaj et al., 2023; Lyu et al., 2024), in contrast to preference feedback considered in our setting. The key idea behind offline reinforcement learning is pessimism, which encourages the learning agent to focus on the policies or actions supported by the dataset (Cheng et al., 2022). However, the gap between learning from reward feedback and preference feedback is currently unclear. Particularly, we find that the batch constrained Q learning (BCQ) algorithm (Fujimoto et al., 2019) shares some similarity with the idea and approach in our work. We highlight the similarity and key difference between PRC and BCQ in the appendix.

**Online PBRL:** Preference-based reinforcement learning is popular in the online learning scenario (Christiano et al., 2017; Ye et al., 2024; Ibarz et al., 2018; Lee et al., 2021b). Various types of preference models and labels have been considered in the tabular case (Fürnkranz et al., 2012; Wirth et al., 2017). Similar to the works that consider the general case, in our work, we consider the preference models to be Bradley-Terry models and output binary preference labels (Ibarz et al., 2018; Lee et al., 2021a). The two-phase learning approach mentioned in Section 1 is prevalent in the online PBRL setting. Note that directly applying online methods to the offline scenario is also likely to be inefficient even when possible (Van Hasselt et al., 2018), as the online learning approaches do not need to be pessimistic.

**Offline PBRL:** Offline preference-based reinforcement learning has attracted much attention recently. Specifically, we find three types of offline PBRL settings attract most attention. 1. Zhan et al. (2023); Zhu et al. (2023) consider the offline learning setting where the training data only consists of preference labels. This setting is similar to the standard reward-based offline RL setting, where the training data only consists of reward labels. This work focuses on this setting. Zhan et al. (2023) proposes an optimization problem following the two-phase learning framework and theoretically proves that the solution to the problem is a near-optimal policy with respect to the training dataset. However, how to solve the optimization problem in practice remains unknown. Concurrently, Zhu et al. (2023) formulates a similar optimization problem under the linear assumption of the environment and proposes a way to solve the problem. Still, the results cannot be extended to the general case without the linear assumption. 2. Sadigh et al. (2017); Shin et al. (2021; 2023) study the PBRL problem in the active learning setting where the agent has access to an offline behavior dataset and can choose a subset of the dataset to be labeled with preference feedback. 3. An et al. (2023); Kim et al. (2023); Hejna & Sadigh (2023); Zhang et al. (2023); Tu et al. (2024); Gao et al. (2024); Kang & Oh (2025); Choi et al. (2024); Liu et al. (2024) consider a different offline learning setting where the agent can access both a dataset of preference labels and an additional trajectory dataset that only contains trajectories. The fundamental difference between this setting and the active learning setting is that, here, the preference labels are given to the trajectories before training. Note that reinforcement learning

from human feedback (RLHF) for natural language processing (NLP) tasks is a special instance of PBRL where humans provide the preference labels and is often studied in the standard offline setting (Ouyang et al., 2022; Bai et al., 2022; Rafailov et al., 2023) which is the same as the setting considered in this work.

## 6   Conclusion and Limitation

In this work, we propose a novel two-step learning algorithm called PRC for the offline PBRL setting. On the standard offline RL benchmarks, we empirically verify that the PRC has high learning efficiency and provide evidence for why it is a more efficient two-step learning algorithm than others. Our framework is limited to the typical offline learning setting where the agent only has access to an offline preference dataset. Our experimental evaluation is limited to continuous control problems, the standard benchmark in RL studies. Our method relies on an external online learning method to learn the policy. Our practical implementation strictly forces the learned policy to output actions in a small action space, which could limit the full potential of the dataset. To approximate the dataset's action distribution, the implementation uses a constrained action space centered at actions given by a learned deterministic imitation policy. This choice is not able to well characterize all possible dataset action distributions. A future extension of this work can be adopting a stronger behavior cloning technique that can better characterize the dataset's action distribution.

### Acknowledgments

We sincerely thank the anonymous reviewers for their insightful comments and helpful suggestions. This work was supported by funding through NSF Grants No. CCF-2238079, CCF-2316233, CNS-2148583 anda Research Gift from Amazon AGI Labs

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

# A    Additional Experiment Results

## A.1    Evaluation on Human Preference Dataset

This section evaluates the performance of our method on the offline PBRL datasets with preference labels collected from real humans. More specifically, we consider the datasets based on the D4RL and MetaWorld benchmark provided in (Kim et al., 2023). The training setup for the D4RL datasets is the same as that of our experiments in Section 4. We follow the training setup for the MetaWorld datasets from (Zhang et al., 2023), which demonstrates high learning efficiency. We adopt the same baselines used in (Zhang et al., 2023), including the RM (based on IQL (Kostrikov et al., 2021)), OPRL (Shin et al., 2023), PT (Kim et al., 2023), IPL (Hejna & Sadigh, 2023), and FTB (Zhang et al., 2023). We use the performances reported by (Zhang et al., 2023) to represent the performance of the baseline methods. Note that the baseline methods here are developed for a different offline PBRL setting. The baseline methods have access to the human preference dataset and a large behavior dataset. Our method has access to the same human preference dataset and the state transition function of the environment.

In Table 2, we show the performance of the policies learned by different methods on the human preference datasets. The results show that PRC outperforms the IPL, PT, and reward modeling methods by a clear margin in most cases. Compared to FTB, PRC has a better performance on both D4RL datasets and the 'plate-slide-v2' dataset, and a worse performance on 'sweep-into-v2'. This is similar to our observations in Table 1 that the two methods have similar performance in general. Our results verify the learning efficiency of our method on the human preference labels and on the MetaWorld benchmark, which is considered more challenging than D4RL.

| Dataset | PRC | RM | OPRL | PT | IPL | FTB |
|---|---|---|---|---|---|---|
| HalfCheetah-Medium-Replay | $\mathbf{41.8 \pm 1.4}$ | 36.0 | 38.3 | $\mathbf{41.1}$ | 36.5 | 38.4 |
| Walker-Medium-Replay | $\mathbf{94.1 \pm 10.9}$ | 75.7 | 64.0 | 79.6 | 8.8 | 79.1 |
| plate-slide-v2 | $\mathbf{0.85 \pm 0.05}$ | 0.00 | 0.44 | 0.45 | 0.38 | 0.51 |
| sweep-into-v2 | $0.35 \pm 0.03$ | 0.00 | 0.21 | 0.18 | 0.14 | $\mathbf{0.97}$ |

Table 2: Comparison between the performance of different learning methods on datasets with human preference feedback. Bolded performances are within 95% of the highest performance. FTB requires much more training time than other methods.

| Dataset | $r = 0.05$ | $r = 0.1$ | $r = 0.2$ | $r = 0.5$ | RM |
|---|---|---|---|---|---|
| HalfCheetah-Medium-Replay | $40.8 \pm 0.2$ | $43.3 \pm 0.2$ | $45.4 \pm 0.5$ | $44.1 \pm 0.7$ | $40.1 \pm 0.7$ |
| Walker-Medium-Replay | $79.5 \pm 8.4$ | $87.9 \pm 6.1$ | $88.6 \pm 2.3$ | $87.3 \pm 5.8$ | $71.8 \pm 6.4$ |

Table 3: Performance of PRC for different values of the constrained action space radius $r$.

## A.2    Ablation Study on Constrained Action Space Radius $r$

This section evaluates the performance of PRC for different values of the constrained action space radius $r$. We test on a wide range of radius coefficients $r \in \{0.05, 0.1, 0.2, 0.5\}$. The other training setups are the same as those of the experiments in Section 4. The results in Table 3 show that PRC can always learn policies with decent performance for different values of $r$, suggesting that PRC is not sensitive to the choice of $r$. PRC achieves a higher performance for $r \in [0.1, 0.5]$. Intuitively, a smaller value of $r$ makes the reinforcement learning process more conservative and less complicated. Therefore, we set $r = 0.1$ for our experiments in Section 4.

## A.3    Baseline Two-step learning algorithms combined with SAC or PPO

In Section 4, we mention that for the two baseline two-step learning PBRL algorithms, using the PPO algorithm for the online RL step learns a better policy compared to using the SAC algorithm. Here, we give some concrete examples in Table 4. For both baseline algorithms, the results show that the performance of

| Dataset | Naive-SAC | Naive-PPO | KL-SAC | KL-PPO |
|---|---|---|---|---|
| HalfCheetah-Medium-Replay | $14.7 \pm 1.2$ | $31.9 \pm 0.8$ | $16.9 \pm 0.6$ | $32.9 \pm 0.9$ |
| Walker2d-Medium-Replay | $22.7 \pm 0.2$ | $49.8 \pm 7.3$ | $25.2 \pm 0.8$ | $45.4 \pm 0.1$ |

Table 4: Baseline Two-step learning algorithms combined with SAC or PPO. 'Naive' is the baseline method that directly optimizes a policy on the learned reward model. 'KL' is the baseline method that applies a KL divergence penalty during policy optimization.

the policy learned by the PPO algorithm exceeds that of the SAC algorithm by a clear margin. Therefore, to avoid underestimating the baseline algorithms, they always use the PPO algorithm for the online RL step.

## B  Proof for Proposition 3.1

*Proof.* For any policy $\pi \in \Pi_C$ and any reference policy $\pi_{\mathrm{ref}} \in \Pi^{\mathcal{S},\mathcal{A}}$, the performance gap between $\pi$ and the learned policy $\hat{\pi}$ satisfies:

$$
\begin{aligned}
R^*(\pi) - R^*(\hat{\pi}) &= \left(R^*(\pi) - R^*(\pi_{\mathrm{ref}})\right) - \left(R^*(\hat{\pi}) - R^*(\pi_{\mathrm{ref}})\right) \\
&= \left\{\left(R^*(\pi) - R^*(\pi_{\mathrm{ref}})\right) - \left(\hat{R}(\pi) - \hat{R}(\pi_{\mathrm{ref}})\right)\right\} + \left(\hat{R}(\pi) - \hat{R}(\pi_{\mathrm{ref}})\right) - \\
&\quad \left\{\left(R^*(\hat{\pi}) - R^*(\pi_{\mathrm{ref}})\right) - \left(\hat{R}(\hat{\pi}) - \hat{R}(\pi_{\mathrm{ref}})\right)\right\} - \left(\hat{R}(\hat{\pi}) - \hat{R}(\pi_{\mathrm{ref}})\right) \\
&\leq \Gamma_{\hat{R}}(\pi, \pi_{\mathrm{ref}}) + \Gamma_{\hat{R}}(\hat{\pi}, \pi_{\mathrm{ref}}) + \hat{R}(\pi) - \hat{R}(\hat{\pi}) \\
&\leq \Gamma_{\hat{R}}(\pi, \pi_{\mathrm{ref}}) + \max_{\pi' \in \Pi_C} \Gamma_{\hat{R}}(\pi', \pi_{\mathrm{ref}}) + \epsilon_{\mathrm{opt}}
\end{aligned}
$$

The first inequality follows the definition of the estimation error $\Gamma$. Since the bound on performance gap holds for any policy $\pi \in \Pi_C$ given any reference policy $\pi_{\mathrm{ref}} \in \Pi^{\mathcal{S},\mathcal{A}}$, the performance of $\hat{\pi}$ can be lower bound by

$$
R^*(\hat{\pi}) \geq \max_{\pi \in \Pi_C, \pi_{\mathrm{ref}} \in \Pi^{\mathcal{S},\mathcal{A}}} \left(R^*(\pi) - \Gamma_{\hat{R}}(\pi, \pi_{\mathrm{ref}}) - \max_{\pi' \in \Pi_C} \Gamma_{\hat{R}}(\pi', \pi_{\mathrm{ref}})\right) - \epsilon_{\mathrm{opt}}
$$

$\square$

## C  Additional Details for Experimental Setups

To learn a reward model, we follow a standard supervised learning framework. Specifically, we use a multilayer perceptron (MLP) structure for the neural network to approximate the utility model. The neural network consists of 3 hidden layers, and each has 64 neurons. We use a tanh function as the output activation so that the output is bound between $[-1, 1]$. To train a deterministic behavior clone policy, the neural network we use to represent the clone policy has the same structure as that for the utility model. To train a stochastic behavior clone policy, the network outputs the mean and standard deviation of a Gaussian distribution separately. The network is an MLP with 3 hidden layers each with 64 neurons. The last layer is split into two for the two outputs. For the mean output, we use a linear function for the last layer. For the standard deviation, we use a linear function and an exp activation for the last layer. In the reinforcement learning step, we use either the SAC algorithm or the PPO algorithm, depending on the dataset. The structure of the neural network for the actor is identical to that of the network used for training a stochastic behavior clone policy. The structure of the neural network for the critic is identical to that of the network used for training the reward model.

## D  Dataset Construction

The construction of our preference dataset is based on the reward-based datasets from the standard D4RL benchmark. We generate synthetic preference labels following the preference model introduced in Section 2 to ensure that the dataset construction is consistent with the problem setting. We formally show the process of data construction as follows.

1. Randomly sample pairs of trajectory clips from the D4RL dataset. Following previous studies (Christiano et al., 2017), the length of the clip is set to be 20 steps.

2. For each pair of trajectory clips, compute the probability of a trajectory to be preferred based on the reward signals. To ensure consistency between different datasets, the reward signals are regularized to be bound in $[-1, 1]$.

3. For each pair of trajectory clips, randomly generate a preference label through a Bernoulli trial with the probability computed above.

4. Return the preference dataset consisting of the trajectory clip pairs and the corresponding preference labels.

# E    Comparison between BCQ and PRC

The PRC algorithm shares several similarities with the famous offline RL algorithm BCQ (Fujimoto et al., 2019) in the reward feedback setting. Both works are based on the common idea of pessimism that constrains the learning agent to policies covered by the dataset. In the practical implementations, both methods adopt the concept of a constrained action space during training. However, here we clarify that PRC and BCQ adopt very different learning strategies.

In our practical implementation, we first train a deterministic imitation policy $\pi^i$. Then we train a policy $\pi(a|s)$ supported on a constrained action space with radius $r$ centered at the action given by $\pi^i$. The goal of our implementation is to find a constrained action space that only includes actions with a high probability of being generated by the behavior policy of the dataset. The purpose of constructing the constrained action space is to exclude the policies not covered by the dataset.

In BCQ, they train a stochastic imitation policy $\pi^i$ and a $Q$ function. Essentially, they want to find a policy $\pi$ that has a similar distribution to $\pi^i$ but is also more likely to select actions of higher value on the $Q$ function. To achieve this, at a state $s$, they randomly sample $n$ responses from the imitation policy $\pi^i(\cdot|s)$, and they want to output the action with the highest value on $Q$. To further enhance the diversity of the samples, they introduce a perturbation function $\epsilon(s, a, \Phi)$ that takes a state $s$ and action $a$ as input, and outputs a perturbation that is in a bound space $\Phi$. In principle, the bounded space $\Phi$ here can be arbitrary and unrelated to the imitation policy, which is completely different from our method. The perturbation function is used to find the action with the highest value on $Q$ in a constrained action space $\Phi$ centered at the sampled actions. The purpose of the constrained action space here is to increase the diversity of the sampled actions. In the end, the BCQ method outputs a policy as $\pi(s) = \arg\max_{a_i + \epsilon(s, a_i, \Phi)} Q(s, a_i + \epsilon(s, a_i, \Phi)), \{a_i \sim \pi^i(\cdot|s)\}_{i=1}^N$. This suggests that BCQ can output any action $a$ with $\pi^i(a|s) > 0$. The pessimism in BCQ comes from sampling actions from the stochastic imitation policy and then outputting an action close to the samples. In this case, the final action distribution is not very different from the imitation policy as long as the number of samples is limited.

In conclusion, BCQ achieves pessimism in a different approach. At a high level, the main purpose of using a constrained action space in our method is to exclude policies not covered by the dataset. In comparison, BCQ uses the bounded action space to enhance the diversity of sampled actions. The principle in the construction of the constrained action space in our method is to include high-probability actions in the behavior policy of the dataset. The bounded action space in BCQ follows no specific principle and can be defined arbitrarily. There are also critical technical differences. 1. Our practical implementation trains a deterministic imitation policy and a standard policy supported on the constrained action space. BCQ trains a stochastic imitation policy and a perturbation 'policy' that takes both state and action as input. 2. Our method learns a policy that can only output actions in a constrained action space with radius $r$ centered at the action given by a deterministic imitation policy. BCQ can output any action that has a positive probability on the distribution given by the stochastic imitation policy. For example, if the imitation policy is a Gaussian policy that has a distribution over the whole action space, then the policy can output any action in the action space. Therefore, our method is very different from BCQ, especially in the usage of the constrained action space.

# F    PRC and FTB Runtime Comparison

For the computational resource, we use a single NVIDIA A100-PCI GPU with 40 GB RAM and an Intel(R) Xeon(R) Silver 4214R CPU @ 2.40GHz with 64 GB RAM. The architecture for all neural networks used in the FTB and PRC algorithms is a multilayer perceptron with 3 hidden layers, and each has 64 neurons. On the datasets we construct, on average, the training time until convergence for the PRC algorithm is $5.14 \pm 1.50$ hours, while for the FTB algorithm it is $34.64 \pm 2.96$ hours.

