# OpenReview forum: "Two-Step Offline Preference-Based Reinforcement Learning on Explicitly Constrained Policies"
_TMLR — Accepted by TMLR_

### Review · Reviewer_yXYF · 2025-07-11

**Summary Of Contributions:**

This paper proposes a method for offline reinforcement learning from preference data. The proposed method constrains the policy so that outputs only the action whose probability in the behavior policy is larger than threshold. In the continuous case, the constraint is within the region whose distance to the output of deterministic behavior policy is lower than threshold. The proposed method demonstrates comparable or better performance with small training time than existing methods.

**Audience:**

No

**Broader Impact Concerns:**

Since this is algorithm paper, I think the risk of ethical problem is small.

**Claims And Evidence:**

No

**Requested Changes:**

Major

Discussion on the technical novelty.

Experimental evaluation on the case of real preference reward and on the case of discrete action.

Experimental comparison to more existing offline methods.

Ablation study on the hyperparameter $r$.


Minor

Proposed method on the continuous space using stochastic behavior policy modeling.

**Strengths And Weaknesses:**

Strength

The author proposes a method for offline reinforcement learning from preference.

The proposed method demonstrates comparable or better accuracy to existing offline methods with lower training computation time.

The author visualizes that the proposed method is robust to the learned reward model, training is stable and efficient.


Weakness

I think the idea of constraining the action space to well-observed actions is naïve and too pessimistic. Doesn’t the proposed method converge fast but the final score is smaller?

Though the author formulates the proposed method with the action probability, the final algorithm in the continuous case does not obey this probabilistic framework and uses distance to the representative deterministic action. For example, we can use probability by modeling behavior policy with gaussian policy.

The experiment is conducted on synthesized preference and on continuous action case. I want to see the results on true preference task and on discrete action case.

Also, I want to see comparison to more existing offline methods (e.g. CQL, TD3+BC, Decision transformer).

Further, I think the performance of the proposed method depends on the constraint parameter $r$. I want to see the ablation study about this parameter.

---

> ### Author Response · Authors · 2025-08-18
> **Response (1)**
>
> Thank you for your constructive comments! The revision can be found in our latest submission. Below are our responses to your concerns.
>
> **Q1:** I think the idea of constraining the action space to well-observed actions is naïve and too pessimistic. Doesn’t the proposed method converge fast but the final score is smaller?
>
> **A1:** Our work focuses on the two-step learning framework as introduced in Section 2.2. Under the framework, by Proposition 3.1, it is necessary to exclude certain policies during the policy optimization process. It is typical in existing offline RL algorithms [1,2,3] to apply policy or action constraints during policy optimization. Our method follows the intuition of constraining the policies that are not covered by the dataset. Furthermore, our key insight is that such policies contribute to both risk of reward over-optimization and reinforcement learning complexity. As we discussed in our response A3 to Reviewer TFpN, the reinforcement learning process on a reward model learned from inaccurate and stochastic binary preference feedback is hard. Therefore, our method explicitly constrains the RL process to only consider policies supported on a constrained action space. This choice decreases the complexity of the RL process, resulting in fewer optimization errors. This is our main difference from previous works and the reason why our method is efficient in practice.
>
> Our empirical results in Table 1 from the paper show that our method learn policies of higher performance compared to other standard methods under the two-step learning framework and the computationally efficient baseline IPL. Therefore, we believe our method is computationally efficient and performs well in practice. It is indeed possible to achieve a better learning efficiency with more computational resources. The FTB method uses the diffusion technique that requires more computational resources, and on some datasets, it learns policies of higher performance than those learned by our method.
>
> [1]: Wu, Yifan, George Tucker, and Ofir Nachum. "Behavior regularized offline reinforcement learning." arXiv preprint arXiv:1911.11361 (2019).
>
> [2]: Kumar, Aviral, et al. "Stabilizing off-policy q-learning via bootstrapping error reduction." Advances in neural information processing systems 32 (2019).
>
> [3]: Fujimoto, Scott, David Meger, and Doina Precup. "Off-policy deep reinforcement learning without exploration." International conference on machine learning. PMLR, 2019.
>
> **Q2:** Though the author formulates the proposed method with the action probability, the final algorithm in the continuous case does not obey this probabilistic framework and uses distance to the representative deterministic action. For example, we can use probability by modeling behavior policy with gaussian policy.
>
> **A2:** If the agent has prior information on the probability distribution of the behavior policy or access to the behavior policy directly, we believe it is a good choice to use the behavior policy directly or learn a stochastic behavior clone policy based on the prior information. However, our experiments focus on datasets from the standard continuous robotic control benchmark, and we have no information about the behavior policy used to collect the dataset. We have tested learning a Gaussian behavior clone policy following the typical approach [1]. In this case, we define the constrained action space to center at the mean of the Gaussian distribution given by the behavior clone policy, and the radius of the action space depends on the variance. However, we find that the center of the distribution is not very close to the dataset action, and the variance is relatively large. As a result, the constrained action space deviates from the dataset actions, and the corresponding reinforcement learning process on the constrained action space fails to learn policies of high true performance.
>
> Therefore, we believe that learning a deterministic behavior clone policy, as in our method, is a more practical and efficient approach in this case. Here, the behavior policy learning process only needs to find the action that is close to the dataset action, which is easier than the Gaussian policy case that trains both a mean and a variance to maximize the probability of the distribution at the dataset action. Such simplicity allows us to define a constrained action space with its center close to the dataset action, and it is easy to learn a high-performing policy from the corresponding reinforcement learning process on this constrained action space.
>
> [1]: Wu, Yifan, George Tucker, and Ofir Nachum. "Behavior regularized offline reinforcement learning." arXiv preprint arXiv:1911.11361 (2019).

---

> > ### Comment · Reviewer_yXYF · 2025-08-19
> >
> > Thank you for the response. As for A2, does not the final algorithm follow the framework in Algorithm 2, since Algorithm 2 assumes stochastic policy? Also, in Algorithm 2, does $\pi_0$ indicate $\pi_b$?

---

> > > ### Author Response · Authors · 2025-08-19
> > >
> > > Thanks for your response!
> > >
> > > In Alg 3, we essentially consider a stochastic behavior clone policy that has a uniform distribution over a constrained action space with a radius $r$. In practice, we fix the value of radius $r$ and only train a deterministic policy $\pi^i$ to generate the center of the constrained action space. The radius $r$ and the deterministic policy $\pi^i$ determine the behavior clone policy $\pi_b$ and the constrained action space. This is also how $pi_b$ and $\pi^i$ are connected. (We improve the notations in the latest revision: $\pi_b$ represents the behavior clone policy, and $\pi^i$ represents a deterministic imitation policy). We make it clear in the revision.
> > >
> > > As we explained in A2, this approach makes the training process simple and the constrained action space centered closer to the behavior policy. While a limitation of the method is that it introduces an additional and critical hyperparameter, $r$, our method is not sensitive to the choice of the value of $r$, as we explained in A4.

---

> > > > ### Comment · Reviewer_yXYF · 2025-08-20
> > > >
> > > > Thank you for the answer. I understand the connection.

---

> ### Author Response · Authors · 2025-08-18
> **Response (2)**
>
> **Q3:** The experiment is conducted on synthesized preference and on continuous action case. I want to see the results on true preference task and on discrete action case.
>
> **A3:** We add experiments for testing our method on datasets of human preferences. In the revision, the experiment results can be found in the appendix. We adopt the standard human preference dataset based on D4RL from [1]. We test on two datasets, ‘HalfCheetah-Medium-Replay’ and ‘Walker2d-Medium-Replay’. The training setup is the same as that of our experiments in the paper. Compared to the results of baseline algorithms given in [2], including FTB, IPL, and RM, our method learns policies of higher performance. The results verify that our method has high learning efficiency when working with true human performance. We also test on a different benchmark, ‘MetaWorld’, with human preference. Our method also achieves high learning efficiency on the Metaworld benchmark. The details can be found in our response A1 to Reviewer duRi. We have included the experiment results in the revision.
>
> To the best of our knowledge, recent PBRL studies mostly focus on robotic tasks with continuous action space, such as D4RL and Metaworld. We could not find a well-established preference dataset that works with the discrete action space. Even though we want to point out that an extension of our approach to the discrete action case is closely related to an efficient learning strategy in RLHF known as best of n sampling. The details can be found in our response A4 to Reviewer duRi.
>
> [1]: Kim, Changyeon, et al. "Preference transformer: Modeling human preferences using transformers for rl." arXiv preprint arXiv:2303.00957 (2023).
>
> [2]: Zhang, Zhilong, et al. "Flow to better: Offline preference-based reinforcement learning via preferred trajectory generation." The Twelfth International Conference on Learning Representations. 2023.
>
>
> **Q4:** Also, I want to see comparison to more existing offline methods (e.g. CQL, TD3+BC, Decision transformer).
>
> **A4:** As mentioned in our response A1 to Reviewer TFpN, in our two-step learning framework, the second step uses online reinforcement learning algorithms (such as SAC) to optimize a policy on the reward model learned from the first step. Most existing offline methods, such as CQL and TD3-BC, are designed for reward-based feedback, which is not directly applicable to our preference-learning setting.
>
> **Q5:** Further, I think the performance of the proposed method depends on the constraint parameter $r$. I want to see the ablation study about this parameter.
>
> **A5:** We add an ablation study on the parameter r in the revision. The results can be found in the appendix. We test on a wide range of radius coefficients $r\in\{0.05,0.1,0.2,0.5\}.$ The other training setups are the same as those of our experiments in the paper. The results show that our method can always learn policies with decent performance for different values of $r$, suggesting that our method is not very sensitive to the choice of $r$. Our method achieves a higher performance when $r\in[0.1,0.5]$. Intuitively, a smaller value of $r$ makes the reinforcement learning process more conservative and less complicated. Therefore, we set $r=0.1$ in practice. We have included the experiment results in the revision.

---

> ### Author Response · Authors · 2025-09-09
>
> Thanks again for your valuable comments!
>
> Please let us know if you still have questions about the claims, evidence, and audience of this work. We are happy to provide additional evidence.

---

### Review · Reviewer_TFpN · 2025-07-24

**Summary Of Contributions:**

This paper proposes an algorithm for offline preference-based reinforcement learning. The algorithm follows a widely adopted two-step approach commonly used in preference learning.
In the first stage, a reward model is trained using a given preference dataset. A policy is then optimized to maximize the expected return as defined by the learned reward model. The main focus of the proposed algorithm lies in the policy optimization during the second stage.
The core idea is to reduce the action space by constraining it to the neighborhood of the behavior policy. Specifically, the policy's action space is restricted around the deterministic behavior policy, and the constrained policy is then optimized using a standard reinforcement learning algorithm.
The paper also empirically demonstrates the challenges of learning from a reward model trained on preference data.

**Audience:**

Yes

**Broader Impact Concerns:**

I do not have any concerns regarding the broader impact of this work.

**Claims And Evidence:**

Yes

**Requested Changes:**

- In Algorithm 2, $\mathcal{L}_{u}$ and $\mathcal{L}_{\pi}$ are introduced before they are properly defined in the main text. It would be advisable to revise the presentation to ensure that these terms are explained beforehand or concurrently for better clarity.

- The authors are encouraged to clarify the rationale for choosing SAC in the offline setting, rather than leveraging more recent offline reinforcement learning algorithms.

- The proposed method appears to share conceptual similarities with Batch-Constrained Q-learning (BCQ) [Fujimoto et al., 2018]. While one difference is the use of a deterministic behavior cloning policy rather than a conditional VAE, a more comprehensive comparison is recommended. If there are additional distinctions between the proposed approach and BCQ, those should be explicitly addressed to clarify the novelty and contributions of this work.

- While the paper is focused on preference-based RL, I do not see what part of the proposed method solves specific challenges in preference-based learning. The contribution of the paper appears to lie in offline policy optimization. Please elaborate this point.

- Although the paper is positioned within the domain of preference-based reinforcement learning, it is unclear which aspects of the proposed method directly address the unique challenges associated with preference-based learning. From the current presentation, the primary contribution appears to lie in offline policy optimization. The authors are encouraged to clarify and elaborate on how the method specifically tackles issues inherent to preference-based settings, such as preference elicitation, reward inference, or robustness to noise in human feedback.

**Strengths And Weaknesses:**

Strengths
- The proposed method exhibits promising empirical performance across the evaluated tasks.
- The core concept is straightforward and grounded in intuitive reasoning.

Weaknesses
- Several sections lack clarity in presentation, which may hinder comprehension.
- The manuscript provides limited discussion of prior work, particularly in relation to closely related approaches.


On page 3, the authors state: “A learning agent is given access to the incomplete MDP of the environment and a dataset.” However, based on the descriptions in Sections 4.1 and 4.2, this does not appear to be accurate. The agent seems to operate in a strictly offline setting, without access to the environment. It would be helpful to clarify whether the policy is trained using online reinforcement learning with the learned reward model, or if it is entirely offline.

On page 6, the manuscript notes that “In practice, we can solve this by applying the SAC or PPO algorithm.” However, it is not clearly specified which algorithm was used in the experiments described in Sections 4.1 and 4.2. Additionally, standard SAC implementations are generally not suitable for offline reinforcement learning settings. Recent offline RL algorithms such as Implicit Q-Learning (IQL) or TD7 may be more appropriate and could enhance reproducibility and performance.

Furthermore, the proposed method shares substantial similarities with Batch-Constrained Q-learning (BCQ) [Fujimoto et al., 2018]. BCQ also involves training a behavior policy and refining it through bounded perturbations. While BCQ employs a conditional variational autoencoder (VAE) to model the behavior policy, the overall framework aligns closely with the method presented in this paper. Given that the primary contribution of the work centers on policy optimization, a more thorough comparison with BCQ, including both conceptual and empirical differences—would strengthen the manuscript.

===comments after rebuttal===

 I think that the authors’ response satisfactorily addressed my concerns. I updated the score accordingly.

---

> ### Author Response · Authors · 2025-08-18
> **Response**
>
> Thank you for your constructive comments! The revision can be found in our latest submission. Below are our responses to your concerns.
>
> **Q1:** It would be helpful to clarify whether the policy is trained using online reinforcement learning with the learned reward model, or if it is entirely offline ... Recent offline RL algorithms such as Implicit Q-Learning (IQL) or TD7 may be more appropriate and could enhance reproducibility and performance
>
> **A1:** We clarify that in our two-step learning framework, the second step uses online reinforcement learning algorithms to optimize a policy on the reward model learned from the first step. Our framework does not use offline RL algorithms for training policies. In the revision, we highlight this point and explicitly show which online RL algorithms we use in our experiments.
>
> **Q2:** Furthermore, the proposed method shares substantial similarities with Batch-Constrained Q-learning (BCQ) [Fujimoto et al., 2018] ... the overall framework aligns closely with the method presented in this paper.
>
> **A2:** We agree that BCQ and many other offline learning algorithms, such as BRAC [1] and BEAR [2], share a similar idea of realizing pessimistic learning as our method. This has also been mentioned by reviewer duRi. The common idea is to constrain the policy optimization process to the policies/actions covered by the dataset. The main consideration in previous works is to penalize the policies that are likely to be overestimated during training. However, the critical insight in our approach is that such policies also contribute to the difficulty of the reinforcement learning process. In our response A3, we discuss why this is an important concern in a preference-based learning setting. Therefore, unlike previous works, including BCQ, that still consider the full policy space with penalties on un-supported polices, our method explicitly excludes the unsupported policies during policy optimization. This is the critical difference between our work and previous work. By explicitly constraining the policies, our method achieves the idea of pessimism while reducing the complexity of the reinforcement learning process. We include the related works mentioned above in the revision.
>
> [1]: Wu, Yifan, George Tucker, and Ofir Nachum. "Behavior regularized offline reinforcement learning." arXiv preprint arXiv:1911.11361 (2019).
>
> [2]: Kumar, Aviral, et al. "Stabilizing off-policy q-learning via bootstrapping error reduction." Advances in neural information processing systems 32 (2019).
>
> **Q3:** While the paper is focused on preference-based RL, I do not see what part of the proposed method solves specific challenges in preference-based learning. The contribution of the paper appears to lie in offline policy optimization. Please elaborate this point.
>
> Although the paper is positioned within the domain of preference-based reinforcement learning, it is unclear which aspects of the proposed method directly address the unique challenges associated with preference-based learning. From the current presentation, the primary contribution appears to lie in offline policy optimization.
>
> **A3:** We believe a challenge in PBRL is that the binary and stochastic preference signals contain less information compared to the exact scalar reward feedback in the standard RL setting. In practice, we show that the policy optimization process on a reward model learned from the preference feedback is hard.
>
> In Section 4.5, we conduct additional experiments to verify this statement. Given the same D4RL dataset, we label the dataset with preference signals and true reward signals. Then, we train two reward models based on the two kinds of signals. Finally, we run the PPO algorithm on the two reward models to train policies. The results in Figure 3 show that when learning on the reward model trained from true rewards, the RL agent can quickly learn some policies that have high performance on the reward model. In comparison, it can take many more training epochs for the agent to find a good policy on the reward model from preference signals, and sometimes, the agent may not be able to find a decent policy after a lot of training epochs.
>
> Our empirical observations suggest that it is harder to learn from a reward model that is trained on preference signals instead of true reward signals. To mitigate the difficulty, our method focuses on reducing the complexity of the reinforcement learning process under the pessimistic learning framework. More details can be found in our response A1 to Reviewer yXYF. We highlight this more in the revision.
>
> **Q4**:In Algorithm 2, $\mathcal{L}{u}\mathcal{L}{\pi}$ are introduced before they are properly defined in the main text. It would be advisable to revise the presentation to ensure that these terms are explained beforehand or concurrently for better clarity.
>
> **A4:** Thanks for the suggestion. We have made the definitions clear in the revision.

---

> > ### Comment · Reviewer_TFpN · 2025-08-20
> >
> > Thank you for the clarification. I understand that the policy is trained using an online RL algorithm.
> > However, I am still unsure why the paper places particular emphasis on the term “offline.” As I see it, the problem setting appears to be the same as in standard RLHF: the reward model is trained from preference data, and then the policy is optimized using this learned reward model.
> >
> > Could you please clarify how this differs from the standard RLHF setup, such as the LLM fine-tuning procedure described in Ouyang et al. (2022)?
> >
> >
> > I would like to ask a few questions to make sure I understand correctly.
> > In Algorithm 3, line 4, what exactly is the function $f$?
> > If the imitation policy is deterministic and given by $\mu(s)$, is it the case that it is given by $\mu(s)+a'$, where $a'$  is generated by the behavior policy?
> >
> > > Therefore, unlike previous works, including BCQ, that still consider the full policy space with penalties on un-supported polices, our method explicitly excludes the unsupported policies during policy optimization.
> >
> > Does “our method explicitly exclude the unsupported policies” refer to optimizing the policy within a radius $r$ around the actions of the imitation policy?
> > If so, note that BCQ also follows the same approach and can likewise be regarded as an approach that excludes the unsupported action space.
> > I would recommend revisiting the BCQ paper. In BCQ, the imitation policy is learned using a VAE, but the policy optimization is still restricted to a limited region around the imitation policy.

---

> > > ### Author Response · Authors · 2025-08-20
> > > **Follow up response (2)**
> > >
> > > **Q7:** Does “our method explicitly exclude the unsupported policies” refer to optimizing the policy within a radius $r$ around the actions of the imitation policy?
> > >
> > > If so, note that BCQ also follows the same approach and can likewise be regarded as an approach that excludes the unsupported action space. I would recommend revisiting the BCQ paper. In BCQ, the imitation policy is learned using a VAE, but the policy optimization is still restricted to a limited region around the imitation policy.
> > >
> > > **A7:** Yes. We agree that BCQ shares a similar format to our practical implementation regarding the constrained action space, but there are big differences between BCQ and our method.
> > >
> > > In our practical implementation, we first train a deterministic imitation policy $\pi^i$. Then we train a policy $\pi(a|s)$ supported on a constrained action space with radius $r$ centered at the action given by $\pi^i$. Note that $\pi^i(s)$ is unique and deterministic for each state $s$, so the constrained action space is deterministic at each state $s$ with a fixed radius $r$. **The goal of our implementation is to find a constrained action space that only includes actions with a high probability of being generated by the behavior policy of the dataset. The purpose of constructing the constrained action space is to exclude the policies not covered by the dataset**.
> > >
> > > In BCQ, they train a stochastic imitation policy $\pi^i$ and a $Q$ function. Essentially, they want to find a policy $\pi$ that has a similar distribution to $\pi^i$ but is also more likely to select actions of higher value on the $Q$ function. To achieve this, at a state $s$, they randomly sample $n$ responses from the imitation policy $\pi^i(\cdot|s)$, and they want to output the action with the highest value on $Q$. To further enhance the diversity of the samples, they introduce a perturbation function $\epsilon(s,a,\Phi)$ that takes a state $s$ and action $a$ as input, and outputs a perturbation that is in a bound space $\Phi$. **Note that the perturbation function takes action as part of the input. In principle, the bounded space $\Phi$ here can be arbitrary and unrelated to the imitation policy, which is completely different from our method**. The perturbation function is used to find the action with the highest value on $Q$ in a constrained action space $\Phi$ centered at the sampled actions.. **Note that the purpose of the constrained action space here is to increase the diversity of the sampled actions.** In the end, the BCQ method outputs a policy as $\pi(s)=\arg\max_{a_i+\epsilon(s,a_i,\Phi)} Q(s,a_i+\epsilon(s,a_i,\Phi)),\{a_i \sim \pi^i(\cdot|s)\}_{i=1}^N.$ **This suggests that BCQ can output any action $a$ with $\pi^i(a|s)>0$. The pessimism in BCQ comes from sampling actions from the stochastic imitation policy and then outputting an action close to the samples. In this case, the final action distribution is not very different from the imitation policy as long as the number of samples is limited.**
> > >
> > > In conclusion, BCQ achieves pessimism in a different approach. **At a high level, the main purpose of using a constrained action space in our method is to exclude policies not covered by the dataset. In comparison, BCQ uses the bounded action space to enhance the diversity of sampled actions.** The principle in the construction of the constrained action space in our method is to include high-probability actions in the behavior policy of the dataset. The bounded action space in BCQ does not need to follow that principle and can be defiincludned arbitrarily. There are also many critical technical differences. 1. Our practical implementation trains a deterministic imitation policy and a standard policy supported on the constrained action space. BCQ trains a stochastic imitation policy and a perturbation 'policy' that takes both state and action as input. 2. Our method learns a policy that can only output actions in a constrained action space with radius $r$ centered at the action given by a deterministic imitation policy. BCQ can output any action that has a positive probability on the distribution given by the stochastic imitation policy. For example, if the imitation policy is a Gaussian policy that has a distribution over the whole action space, then the policy can output any action in the action space. Therefore, our method is very different from BCQ, especially in the usage of the constrained action space.
> > >
> > > Thanks for the question. We find it important to clarify the difference between BCQ and our work, and we have added the clarification to the appendix in the revision.

---

> > > > ### Comment · Reviewer_TFpN · 2025-08-24
> > > > **requests for revisions**
> > > >
> > > > Thank you for your response. Based on your previous replies, I have a few requests for revisions:
> > > >
> > > > - The current title, "Two-Step Offline Preference-Based Reinforcement Learning", is somewhat ambiguous. As it stands, it may give the impression that the entire learning process is offline, which could lead to misunderstandings about the content of the paper. I believe it would be more accurate to change it. For example, something like "Two-Step Preference-Based Reinforcement Learning with Offline Reward Modeling" may better reflect the actual content.
> > > >
> > > > - In Algorithm 3: PRC practical implementation, I suggest avoiding the use of the mapping function $f$ and instead expressing the update more explicitly, such as $a = \mu(s) + a'$ where $a' \sim \pi'(a|s)$. Despite being labeled a "practical implementation," the explanation of the mapping function $f$ is vague and unclear, making it difficult to understand.
> > > >
> > > > - Please discuss the differences from BCQ clearly and accurately. In the proposed method, the parameter $r$ is ultimately a hyperparameter, just as $\phi$ is in BCQ. Moreover, both methods involve learning actions in the form of $a = \mu(s) + a'$. Please ensure your description acknowledges these similarities. Additionally, I believe the statement “BCQ uses the bounded action space to enhance the diversity of sampled actions” is inaccurate. In BCQ, the bounded action space is used to prevent the generation of out-of-distribution actions during offline RL training. In that sense, the reason for using a bounded action space is quite similar to that of the proposed method.
> > > >
> > > > - I have also noticed a potential logical flaw in the proposed method. In your approach, a deterministic policy is used to learn $\pi^i$, and the neighborhood around it defined by radius $r$ is treated as the region where $\pi^i(a|s) > p$. However, this assumption does not hold when the action distribution in the dataset is multimodal. For example, consider a case where the action space is in the range [−1,1], and the dataset contains actions uniformly distributed in the intervals [−0.3,−0.1] and [0.1,0.3]. In such a case, $\pi^i(a|s)$ would likely output 0, but the density around 0 is clearly zero. If this observation is correct, then it should be explicitly stated that the method fails to capture the valid action region properly when the action distribution in the dataset is multimodal.
> > > >
> > > > - Regarding the experiments, you mentioned that PPO is used for the baseline two-step learning methods. Just to confirm, are both Naive Two-Step and KL Two-Step using PPO? You also mentioned that the naive two-step method with PPO performed better than the naive step with SAC. However, since the proposed method uses SAC, results using SAC for the naive two-step baseline should also be presented. Moreover, I suspect the poor performance of naive two-step with SAC stems from the reward model failing to learn the correct rewards. Therefore, in addition to the current learning curves using the true reward, please also present learning curves evaluated using the learned reward model. In this case, I expect that even if the return calculated from the learned reward model increases during training with SAC for naive two-step, the true return may not improve. If such results are observed, it would highlight the issues of learning with a learned reward model more clearly and in turn better demonstrate the effectiveness of the proposed method.

---

> > > > > ### Author Response · Authors · 2025-08-29
> > > > > **Follow-Up Response (1)**
> > > > >
> > > > > Thanks for your response! We truly appreciate your thoughtful comments. They greatly improve the quality and presentation of this study.
> > > > >
> > > > > We have updated the revision according to your valuable suggestions. Below is our response to the comment.
> > > > >
> > > > > **Q1:** The current title, "Two-Step Offline Preference-Based Reinforcement Learning", is somewhat ambiguous.
> > > > >
> > > > > **A1:** Thanks for the suggestion. We rename our work as “Preference-Based Reinforcement Learning with Offline Reward Modeling on Explicitly Constrained Policies” and modify related text in the revision.
> > > > >
> > > > > **Q2:** In Algorithm 3: PRC practical implementation, I suggest avoiding the use of the mapping function f  and instead expressing the update more explicitly,
> > > > >
> > > > > **A2:** Thanks for the suggestions. We have modified Algorithm 3 and its explanations accordingly in the revision.
> > > > >
> > > > > **Q3:** Please discuss the differences from BCQ clearly and accurately.
> > > > >
> > > > > **A3:** We acknowledge the similarity in the revision that both BCQ and our method use actions in the format of $a=\mu(s)+a’$, and the output of the perturbation actions in BCQ is in a constrained space. However, we want to clarify that BCQ and we use the constrained action space for different purposes. We quote from the third paragraph on page 6 in the [BCQ](http://proceedings.mlr.press/v97/fujimoto19a/fujimoto19a.pdf) paper:  “To increase the diversity of seen actions, we introduce a perturbation model ξφ(s, a, Φ), which outputs an adjustment to an action a in the range [−Φ, Φ].” In addition, while the perturbation is constrained, the action output by BCQ is not restricted to a constrained action space. This is because BCQ samples actions to perturb from the behavior policy, which is a stochastic policy that, in principle, can be supported on the whole action space. We utilize the constrained action space to exclude policies unsupported by the dataset, which differs from BCQ. We believe BCQ achieves pessimism by sampling multiple actions from the behavior policy (the VAE model) and then outputting the action with the best value. In this case, the resulting distribution of the output actions would not be very different from the distribution given by the behavior policy, which intuitively realizes the idea of pessimistic learning.
> > > > >
> > > > > **Q4:** I have also noticed a potential logical flaw in the proposed method.
> > > > >
> > > > > **A4:**
> > > > > We clarify that in our empirical evaluation, the Medium-Expert dataset is multimodal. Half of the trajectories in the dataset are collected by a medium policy, while the rest are collected by an expert policy. The empirical results verify that our method can learn policies of high performance in these multimodal datasets. The reason could be that at each state, the behavior clone policy learns to predict an action that is close to the action given by either policy, instead of some actions in the middle. Another factor is that the state distribution induced by the two policies can be different. For a state in the dataset, one policy may have a high probability of visiting it, while the other has a low probability. So at each state, the action distribution of one policy matters more than the other.
> > > > >
> > > > > In principle, the general framework we introduce in Algorithm 1 can handle datasets with different kinds of action distributions. The idea of Algorithm 1  is to approximate the dataset’s action distribution (by learning a behavior clone policy), and then explicitly exclude the actions with low probability during policy optimization. Due to the limitation on the dataset’s size and the complexity of the problem, it is challenging to well approximate the dataset’s action distribution in practice. A standard way in DRL to learn a stochastic distribution over the action space is to train a Gaussian distribution. However, a Gaussian distribution can not characterize all possible distributions. In practice, we also find that the learned Gaussian distribution cannot well predict the action in the dataset, which could be a result of the complexity of the learning process. Implementing Algorithm 1 based on the Gaussian behavior clone policy has low performance in practice. Therefore, in our practical implementation, we consider a simple behavior clone policy that generates a uniform distribution over a constrained action space. The simplicity of the behavior clone policy enables it to accurately predict actions close to the ones in the dataset, and our method achieves high empirical performance.
> > > > >
> > > > > We acknowledge the limitation of this approach that the behavior clone policy in our implementation cannot characterize all kinds of dataset action distribution. A future extension of this work is to adopt a stronger behavior cloning technique that can better characterize the dataset’s action distribution. We discuss the limitations of our practical implementation and the future directions for our method in the revision.

---

> ### Author Response · Authors · 2025-08-20
> **Follow up response (1)**
>
> Thanks for your response! Below are our responses to the new concerns.
>
> **Q5:** However, I am still unsure why the paper places particular emphasis on the term “offline.” As I see it, the problem setting appears to be the same as in standard RLHF: the reward model is trained from preference data, and then the policy is optimized using this learned reward model.
>
> **A5:** The offline PBRL setting considered in this work learn from offline preference signals collected before the training process. Therefore, we emphasize "offline" in our setting.
>
> The typical RLHF setting, such as the ones studied in [1,2], can be considered as a special case of the setting we consider. They work in an offline setting with an offline preference dataset collected before training. [3,4] provide a clear explanation of the offline RLHF setting and distinguish it from other RLHF settings. The main difference is that RLHF considers a bandit setting with a discrete action space, and there is no state transition or long-term reward optimization. In our response [A4](https://openreview.net/forum?id=LxPg5GJuY3&noteId=a037CDdab3) to Reviewer duRI, we explain that a popular algorithm, 'best of n sampling' in RLHF, can be considered as an extension of our method to RLHF.
>
> [1]: Liu, Zhihan, et al. "Provably mitigating overoptimization in rlhf: Your sft loss is implicitly an adversarial regularizer." Advances in Neural Information Processing Systems 37 (2024): 138663-138697.
>
> [2]: Huang, Audrey, et al. "Correcting the mythos of kl-regularization: Direct alignment without overoptimization via chi-squared preference optimization." arXiv preprint arXiv:2407.13399 (2024).
>
> [3]: Dong, Hanze, et al. "Rlhf workflow: From reward modeling to online rlhf." arXiv preprint arXiv:2405.07863 (2024).
>
> [4]: Xiong, Wei, et al. "Iterative preference learning from human feedback: Bridging theory and practice for rlhf under kl-constraint." arXiv preprint arXiv:2312.11456 (2023).
>
>
> **Q6:**  In Algorithm 3, line 4, what exactly is the function $f$? If the imitation policy is deterministic and given by $\mu(s)$, is it the case that it is given by $\mu(s)+a'$, where $a'$ is generated by the behavior policy?
>
> **A6:** $f$ is action mapping function. We think the understanding 'If the imitation policy is ... generated by the behavior policy?' is correct. For clarity, we explain the function $f$ using the notations in this work. The function maps the action generated by the current policy on the constrained action space $a'=\pi(s)$ to the action in the actual action space $a=\pi^i(s) + a'$, where $\pi^i$ is the learned deterministic imitation policy (we omit the projection here for simplicity).

---

> ### Author Response · Authors · 2025-08-29
> **Follow-Up Response (2)**
>
> **Q5:** Regarding the experiments, you mentioned that PPO is used for the baseline two-step learning methods. Just to confirm, are both Naive Two-Step and KL Two-Step using PPO? Moreover, I suspect the poor performance of naive two-step with SAC stems from the reward model failing to learn the correct rewards.
>
> **A5:** We confirm that both baseline two-step learning algorithms use PPO. We report the results based on PPO to avoid underestimating the baselines. In the revision, we show some example results when the baseline methods use SAC. We show that for both baseline two-step learning algorithms, the performance of the policy learned by PPO (used for the online RL step) exceeds that of the SAC by a clear margin. While PPO achieves a better learning result, its performance is still relatively low compared to our method. We agree that the reason mentioned in the comment causes the failure of the traditional method. In fact, we have already reported the results for the learning curve on both true performance and simulated performance on the learned reward model in Figure 1. The results show that for the PRC algorithm, during training, the trend of the learned policies’ performance measured on the learned reward model is aligned with that measured on the true reward. In contrast, for the KL-regularized two-step learning method, we find multiple cases where the trends can even be opposite when evaluated on the reward model and on the true reward. During training, the simulated performance of the learned policy increases while the actual performance decreases. This is a result of relying on overestimated rewards during policy optimization. Since the methods use the same learned reward model, the results support that our method achieves pessimistic learning effectively by exploiting the more accurate prediction of the learned reward model. In the revision, we add additional explanation to highlight the above statement more clearly.

---

> > ### Comment · Reviewer_TFpN · 2025-09-03
> >
> > Thank you for the clarification. I feel that the authors’ response satisfactorily addressed my concerns, and I am grateful for their efforts to accommodate my suggested revisions.

---

> > > ### Author Response · Authors · 2025-09-09
> > >
> > > Thanks so much for your acknowledgement of our rebuttal!
> > >
> > > Please let us know if you still have questions about the claims, evidence, and audience of this work. We are happy to provide additional evidence.

---

> > > ### Author Response · Authors · 2025-09-12
> > >
> > > Thanks again for the wonderful discussion. We are very happy that our response satisfactorily addressed your concerns. However, we notice that your rating for **'claim and evidence'** is still **'No'**. Is there any remaining issue? We are happy to address further concerns about the claim and evidence

---

> > > > ### Comment · Reviewer_TFpN · 2025-09-13
> > > >
> > > > Thank you for letting me know. As far as I understand, the authors can see the initial score, but the final recommendation is made based on the updated score. Therefore, it should not cause any problems. I have also updated the initial review just to be safe.

---

> > > > > ### Author Response · Authors · 2025-09-13
> > > > >
> > > > > Thank you so much for kindly sharing the information with us and your update!

---

### Review · Reviewer_duRi · 2025-08-04

**Summary Of Contributions:**

The paper focuses on Preference-Based Reinforcement Learning and introduces a simple formulation that constrains the action space by excluding out-of-distribution state action pairs. Experimental results on standard continuous-control benchmarks demonstrate that this approach improves performance and sample efficiency compared to existing baselines.

**Audience:**

Yes

**Broader Impact Concerns:**

Broader impact concerns are related to the use of preference-based learning in the context of training large models. Although I believe there are no direct concerns related to the proposed method, I would have appreciated a Broader impact statement.

**Claims And Evidence:**

Yes

**Requested Changes:**

Please, refer to the Weaknesses.

**Strengths And Weaknesses:**

**Strengths**

1. Simplicity: The proposed method is simple, easy to implement, and addressing issues in preference-based offline reinforcement learning.

2. Theoretical Justification: The authors provide theoretical support showing that constraining the policy to stay within high-probability regions reduces both reward estimation and optimization errors.

**Weaknesses**

1. While the idea of constraining policies to high-probability actions is well-established in Offline RL [1], the main contribution lies in applying it to the preference-based setting. However, the method is evaluated solely on D4RL tasks, a benchmark that is increasingly considered outdated even for standard Offline RL. This limits the impact and relevance of the empirical validation.

2. The approach assumes a reliable behavior cloning policy to define the constrained action set, which may be difficult to obtain in practice, especially in noisy or low-data regimes.

3. The paper briefly mentions its limitations but does not thoroughly analyze failure modes or potential drawbacks of the proposed method.

4. It is uncertain whether this simple approach can generalize to complex real-world scenarios, such as training chatbots or other industrial applications. The authors should discuss the broader applicability and potential extensions.

**References**

[1] Kumar A, Fu J, Soh M, Tucker G, Levine S. Stabilizing off-policy q-learning via bootstrapping error reduction. Advances in neural information processing systems. 2019;32.

---

> ### Author Response · Authors · 2025-08-18
> **Response (1)**
>
> Thank you for your constructive comments! The revision can be found in our latest submission. Below are our responses to your concerns.
>
> **Q1:** However, the method is evaluated solely on D4RL tasks, a benchmark that is increasingly considered outdated even for standard Offline RL. This limits the impact and relevance of the empirical validation
>
> **A1:** We believe that D4RL remains an important benchmark for offline RL, as many recent RL studies, such as [1, 2, 3, 4], utilize the D4RL benchmark for evaluation. The continuous robotic control environments in D4RL have been playing critical roles in deep RL research. Testing on more recent datasets can provide a more comprehensive evaluation of our method. Therefore, we test our method on a recent RL benchmark, “MetaWorld”.
>
> We follow the standard training setup in [4] and test our method on two datasets, ‘plate-slide-v2’ and ‘sweep-into-v2’. We have included the experiment results in the appendix (in the revision). Based on the baseline results given in [4], our methods outperform the IPL, PT, and reward modeling methods. Compared to FTB, our method performs better on ‘plate’ and worse on ‘sweep’. This is similar to our observation on the D4RL dataset. Our results verify the learning efficiency of our method on the more recent and challenging dataset ‘MetaWorld’.
>
> [1]: ​​Tu, Songjun, et al. "In-dataset trajectory return regularization for offline preference-based reinforcement learning." Proceedings of the AAAI Conference on Artificial Intelligence. Vol. 39. No. 20. 2025.
>
> [2]: Liu, Xiao-Yin, et al. "LEASE: Offline Preference-based Reinforcement Learning with High Sample Efficiency." arXiv preprint arXiv:2412.21001 (2024).
>
> [3]: Hejna, Joey, and Dorsa Sadigh. "Inverse preference learning: Preference-based rl without a reward function." Advances in Neural Information Processing Systems 36 (2023): 18806-18827.
>
> [4]: Zhang, Zhilong, et al. "Flow to better: Offline preference-based reinforcement learning via preferred trajectory generation." The Twelfth International Conference on Learning Representations. 2023.
>
> **Q2:** The approach assumes a reliable behavior cloning policy to define the constrained action set, which may be difficult to obtain in practice
>
> **A2:** It is typical in offline learning algorithms to learn a behavior clone policy from the dataset [1,2,3]. In a related topic, reinforcement learning from human feedback (RLHF), learning a behavior policy from the dataset as the starting point of the RLHF training process has already become a convention [4]. We believe this is a practical assumption because learning a behavior policy only requires supervised learning, which is simpler than RL. Therefore, when training on an RL dataset, it is usually feasible to run supervised learning to train a behavior clone policy. We include this assumption as a limitation in the revision.
>
> [1]: Wu, Yifan, George Tucker, and Ofir Nachum. "Behavior regularized offline reinforcement learning." arXiv preprint arXiv:1911.11361 (2019).
>
> [2]: Kumar, Aviral, et al. "Stabilizing off-policy q-learning via bootstrapping error reduction." Advances in neural information processing systems 32 (2019).
>
> [3]: Fujimoto, Scott, David Meger, and Doina Precup. "Off-policy deep reinforcement learning without exploration." International conference on machine learning. PMLR, 2019.
>
> [4]: Stiennon, Nisan, et al. "Learning to summarize with human feedback." Advances in neural information processing systems 33 (2020): 3008-3021.
>
>
> **Q3:** The paper briefly mentions its limitations but does not thoroughly analyze failure modes or potential drawbacks of the proposed method
>
> **A3:** In the revision, we include more discussion on the drawbacks of our method in the limitations. The additional limitations include 1. Our method strictly forces the learned policy to a constrained policy space, which could limit the full potential of the dataset. 2. Our method relies on an external online learning method to learn the policy 3. Our method assumes it can learn a reliable behavior policy that can represent the dataset distribution well.

---

> ### Author Response · Authors · 2025-08-18
> **Response (2)**
>
> **Q4:** It is uncertain whether this simple approach can generalize to complex real-world scenarios, such as training chatbots or other industrial applications. The authors should discuss the broader applicability and potential extensions
>
> **A4:** We believe our method can be generalized to reinforcement learning from human feedback (RLHF). RLHF focuses on the problem of aligning large language models to human preferences, which has important real-world applications. The problem is essentially a bandit setting [1] with a discrete action space that has no state transition or long-term reward. Given a preference dataset, one can first learn a behavior clone policy and a reward model. Following the idea of constraining low probability actions in our method, one can focus on the actions that have a high probability of being sampled from the behavior policy, and then output the action that has the highest reward on the learned reward model. This is closely related to an existing efficient RLHF approach, best of n sampling [2].
>
> The best of n sampling process also learns a behavior clone policy and a reward model from the dataset first. Then it uses a generation method, such as top-k, to sample several high-probability actions from the behavior policy, which is similar to constraining the low-probability actions in our method. At last, it outputs the sampled action with the highest reward on the reward model, which is similar to finding the optimal action on the reward function in the constrained action space in our method.
>
> Therefore, the best of n sampling method can be considered as one practical implementation of our method for RLHF. We believe our findings in this work can motivate more efficient variants of the best of n sampling approaches
>
> [1]: Liu, Zhihan, et al. "Provably mitigating overoptimization in rlhf: Your sft loss is implicitly an adversarial regularizer." Advances in Neural Information Processing Systems 37 (2024): 138663-138697.
>
> [2]: Beirami, Ahmad, et al. "Theoretical guarantees on the best-of-n alignment policy." arXiv preprint arXiv:2401.01879 (2024).

---

> ### Author Response · Authors · 2025-09-09
>
> Dear Reviewer, thanks again for your valuable comments! Please let us know if you still have questions about the claims, evidence, and audience of this work. We are happy to provide additional evidence.

---

### Author Response · Authors · 2025-10-24

We have uploaded the camera-ready version and ensured that all the suggestions and requested changes are adequately included. We sincerely thank all the reviewers and the action editor for your insightful comments and suggestions that greatly helped us improve this work, and we formally acknowledge your contribution in the camera-ready version.

Thanks to all of you again for the great discussion during the rebuttal.

---

### Decision · Action_Editor_GMbD · 2025-10-01

**Recommendation:** Accept with minor revision

**Additional Comments:**

The authors have already addressed several of the reviewers' concerns during the discussion period. Nevertheless, the authors should:
* Extend the discussion regarding the relation to BCQ in the main paper (there was a long discussion here and the key conclusions should make it into the main paper).
* Revise the paper to ensure that all raised ambiguities have been resolved.
* Make sure that the related work section covers the works brought up during the review process.
* Ensure a sufficient discussion of the limitations of the proposed approach and its evaluation, also regarding the assessment of generalization.

**Audience:**

Yes

**Audience Explanation:**

While the proposed approach shares similarities with existing works, there are subtle differences. The improved performance over the considered baselines on a relevant problem warrants relevance to individuals in TMLR's audience.

**Claims And Evidence:**

Yes

**Claims Explanation:**

The authors demonstrate the effectiveness of their proposed approach (PRC) in comparison to baselines in several experiments on robot control environments.

---

> ### Author Response · Authors · 2025-11-07
>
> Dear Action Editor, Thank you again for your suggestions and acknowledgment of this work. We have double-checked the camera-ready submission and believe that all comments and discussions during rebuttal are addressed. Besides the changes we have already made during the rebuttal, we ensure that the latest request changes have also been properly addressed:
>
> 1. We provide a detailed comparison between BCQ and PRC in Appendix E.
>
> 2. We revise all the claims and theorems to ensure no ambiguity.
>
> 3. We cite the works mentioned during the discussion in the related work section.
>
> 4. We have listed all the limitations mentioned during the discussion in the conclusion section.
>
> 5. We have provided and discussed the experimental result that generalizes our method to more challenging environments (MetaWorld) and real human preferences in Appendix A.
>
> Thanks again to you and all reviewers for your help. Please let us know if there are any other issues we need to address in the camera-ready.

---

> > ### Comment · Action_Editor_GMbD · 2025-11-09
> >
> > Thank you for your updates.
> >
> > In the current version of the camera ready paper the link to OpenReview links to the discussion of a different paper. Please fix this.

---

> > > ### Author Response · Authors · 2025-11-09
> > >
> > > Sorry for the mistake. We have fixed the link in the latest camera-ready version.